# Use of Domain Labels during Pre-Training for Domain-Independent WiFi-CSI Gesture Recognition

**DOI:** 10.3390/s23229233

**Published:** 2023-11-16

**Authors:** Bram van Berlo, Richard Verhoeven, Nirvana Meratnia

**Affiliations:** Department of Mathematics and Computer Science, Eindhoven University of Technology, P.O. Box 513, 5600 MB Eindhoven, The Netherlands; p.h.f.m.verhoeven@tue.nl (R.V.);

**Keywords:** domain shift, domain-independent self-supervised learning, adversarial classification, WiFi-CSI, gesture recognition

## Abstract

To minimize dependency on the availability of data labels, some WiFi-CSI based-gesture recognition solutions utilize an unsupervised representation learning phase prior to fine-tuning downstream task classifiers. In this case, however, the overall performance of the solution is negatively affected by domain factors present in the WiFi-CSI data used by the pre-training models. To reduce this negative effect, we propose an integration of the adversarial domain classifier in the pre-training phase. We consider this as an effective step towards automatic domain discovery during pre-training. We also experiment with multi-class and label versions of domain classification to improve situations, in which integrating a multi-class and single label-based domain classifier during pre-training fails to reduce the negative impact domain factors have on overall solution performance. For our extensive random and leave-out domain factor cross-validation experiments, we utilise (i) an end-to-end and unsupervised representation learning baseline, (ii) integration of both single- and multi-label domain classification, and (iii) so-called domain-aware versions of the aformentioned unsupervised representation learning baseline in (i) with two different datasets, i.e., Widar3 and SignFi. We also consider an input sample type that generalizes, in terms of overall solution performance, to both aforementioned datasets. Experiment results with the Widar3 dataset indicate that multi-label domain classification reduces domain shift in position (1.2% mean metric improvement and 0.5% variance increase) and orientation (0.4% mean metric improvement and 1.0% variance decrease) in domain factor leave-out cross-validation experiments. The results also indicate that domain shift reduction, when considering single- or multi-label domain classification during pre-training, is negatively impacted when a large proportion of negative view combinations contain views that originate from different domains within a substantial amount of mini-batches considered during pre-training. This is caused by the view contrastive loss repelling the aforementioned negative view combinations, eventually causing more domain shift in the intermediate feature space of the overall solution.

## 1. Introduction

Gesture recognition is important for enabling a variety of Human Computer Interaction (HCI) applications such as intuitive clinical device interaction during surgery, vehicle infotainment systems, and automatic sign language translation [1,2]. Recently, gesture recognition based on Wireless Fidelity (WiFi)-Channel State Information (CSI) data has gained popularity due to its unobtrusive sensing nature. Raw WiFi-CSI data contains amplitude attenuation and phase shift changes of WiFi signals between transmitter(s) and receiver(s) across time [3]. Different gestures cause signals to be absorbed, diffracted, etc., along different simultaneous paths, resulting in different amplitude attenuation and phase shift patterns measured at the receiver.

Gesture recognition with WiFi-CSI data can be conducted using deep learning, most of which use supervised learning [4,5,6], which relies on the availability of a vast amount of training data and task specific labels. Generating task specific labels is labor intensive and is typically conducted by people who have a deep understanding of the tasks being performed. Using application users for labeling requires the seamless integration of a labeling task with the application to prevent immersion loss and is not always ideal due to actions performed under stress, fatigue, or with malicious intent. Therefore, many researchers have focused on unsupervised pre-training approaches [7,8,9,10], with which after pre-training, a downstream task model can be created and trained with a small labeled training dataset.

A problem with standard unsupervised pre-training approaches is that structural information learned from unlabeled data is influenced by domain factors present during data collection. WiFi-CSI domain factor examples include sensing environment, object/person characteristics, and position/orientation with respect to the sensing system [6]. Domain factors cause feature distribution shifts in the data used for training and inference phases. This leads to a performance degradation of the inference phase [11]. In unsupervised learning, the inference phase refers to the phase after fine-tuning a downstream classifier and using that classifier to make predictions. Geometric learning [10] has proven to be successful in reducing the impact of domain factors. It, however, depends on domain factor knowledge encoded in a deep neural network used as an inductive bias. Domain adaptation by means of an extra Maximum Mean Discrepancy (MMD) loss [7], on the other hand, requires additional training every time a new domain (unique combination of domain factors) appears.

In this paper, we address the influence of domain factors in the pre-training phase and propose to integrate a domain-independent feature extractor learned by means of an adversarial domain classification [12] in it. Even though this brings labeling back to the unsupervised pre-training phase, it is an initial step towards the automatic discovery of domains during the pre-training phase. Our proposed method differs from previous work on domain-independent feature extraction such as [12,13,14], as they only consider adversarial domain classification with a supervised deep learning approach. The method was created with two hypotheses in mind. (i) “*The influence of domain factors can be mitigated in a downstream task classifier by introducing adversarial domain classification in a deep neural network used in the unsupervised pre-training phase*”. (ii) “*In situations where the downstream task classifier performance does not improve, or marginally improves, due to the multi-class and single label domain classifier used during pre-training failing to reduce the negative performance impact of domain factors, a multi-class multi-label domain classifier can be introduced instead. The multi-class multi-label domain classifier, during pre-training, does cause the eventual downstream task classifier performance to improve because the negative performance impact of domain factors is reduced. The multi-class multi-label domain classifier exploits the positive and negative view combination matrix created in the Normalized Temperature-scaled cross-entropy (NT-Xent) loss function during pre-training*”. Our unsupervised pre-training approach is inspired by the contrastive unsupervised pre-training approach proposed by Lau et al.  [8]. It is, however, different because it considers (i) adversarial domain classification and (ii) sharing the feature extraction pipeline block between views in a specific view combination in the pre-training phase. The contributions of this paper are:Integration of adversarial domain classification in the pre-training phase of a self-supervised contrastive deep learning-based approach.Introduction of a multi-class multi-label domain classification loss capable of improving adversarial domain classification performance in situations, in which standard multi-class and single label domain classification loss functions fail.Demonstrate that adversarial domain classification integration is negatively impacted by a large proportion of negative view combinations with views that originate from different domains within a substantial amount of positive view combination mini-batches considered during pre-training.Present clear future research directions for domain generalization in the area of unsupervised representation learning with WiFi-CSI data.

## 2. Related Work

### 2.1. Unsupervised Representation Learning at Different Label Distributions

Unsupervised representation techniques for WiFi CSI data are either based on few-shot learning [15] or contrastive learning [16]. In few-shot learning-based approaches, a contrastive pre-training task is followed by *n*-shot (where *n* is number of samples per class) and *m*-way (where *m* is the total number of classes considered) classification tasks based on a representation distance comparison. This is helpful in situations in which the downstream dataset only contains a few samples per class [15]. In contrastive pre-training, structural information is learned by intermediate feature distance maximization between samples belonging to different classes and distance minimization for similar classes. Labels are required for indicating where distance should be maximized or minimized. Afterwards, the classification task typically generalizes well to unseen task labels.

For *n*-shot gesture classification, Hu et al. [9] used MobileNet [17] convolutional and squeeze/excitation blocks as a backbone network. Yang et al. [7] used a Siamese network with two separated CNN + Bidirectional Long Short-Term Memory (BiLSTM) network twins that took two gesture input samples and minimized the pairwise loss. The twins mitigated data feature distribution shift by bringing the source and target probability distributions closer together via minimizing an empirical estimation of the MMD between distributions in a Reproducing Kernel Hilbert Space (RKHS).

Contrastive learning, during pre-training, tries to learn structural information in earlier feature extractor representations that are relevant for a variety of downstream tasks by comparing positive against negative input sample combinations. Combination creation can utilize task labels to group together input samples belonging to similar/different classes. However, this is not required since positive combinations can also be created based on an augmented and unaugmented version of an input sample or subsections of an input sample. Combinations of the first element of a positive combination and any other element are assumed to be negative. A downside to contrastive learning is that it always has to fine-tune a downstream task classifier using a low labeled data volume prior to the obtained structural information during pre-training being useful for a specific task. Fine-tuning a downstream classifier works well when the downstream dataset contains a medium to high number of samples (i.e., 40–60 or more) per class  [18]. During fine-tuning, feature extraction backbone weights are either frozen and the backbone is integrated with the classifier, or features are extracted from the backbone prior to classifier training.

Lau et al. [8] considered contrastive pre-training without integration of domain shift preventing techniques and with two spectrogram views. Each spectrogram pair contained spectrograms denoting the same class captured under the same domain factors except receiver position and the other pairs contain spectrograms for different classes. They used Normalized Temperature-scaled cross-entropy (NT-Xent) [19] contrastive loss. Xu et al. [20], instead of views, took spectrograms augmented by, among other augmentation methods, time direction flipping and Sobel operator processing. Pre-training considered two data flows per input sample, one for time- and one for channel-wise augmentations. The backbone was made of custom convolution-augmented transformer blocks. The contrastive loss considered was also NT-Xent. Jianfei et al. [10] augmented spectrograms with Gaussian noise. The contrastive loss considered was a consistency loss using Kullback-Leibler (KL) divergence in dual form. The influence of the domain shift was mitigated by encoding geometric information in the deep neural network by means of geometric embeddings. The consistent geometric structure was learned by minimizing KL [21] divergence between the embeddings together with minimizing the contrastive loss.

### 2.2. Related Work on Domain-Independent Learning with Adversarial Domain Classification

Domain-independent learning by means of an adversarial domain classification steers model training towards giving priority to feature commonness across domain factors [12]. During training, besides having a task output that produces class predictions, the network also has a ’discriminator’ output that produces domain predictions. An adversarial training procedure is used, where both outputs play a min-max game.

Current adversarial domain classification [12,13,14] techniques rely on the availability of domain labels. This is a technique caveat, since labeling is a labor-intesive task often requiring the involvement of task experts for producing the labels. However, an effective step towards manual labeling in the future could be an automatic domain discovery. Another usage caveat is that the adversarial domain classification requires model re-training when additional domain factors, rather than domains, arise during the inference time [6,22]. This type of retraining requirement is more practical than the retraining requirement when considering domain adaptation, since adaptation also requires retraining when encountering new domains.

Jiang et al. [12] proposed a balance regularization technique, which added loss based on Jensen–Shannon divergence [23] between the output probability distribution and an auxiliary distribution derived thereof. The auxiliary distribution for every class was created by normalizing the output probability by the total number of class predictions with a similar domain label compared to the input domain label. This regularization technique improved the eventual inference performance after adversarial training.

## 3. Methodology

Our proposed pipeline integrating adversarial domain classification in the pre-training phase is illustrated in Figure 1. Here, we explain different building blocks of the pipeline. Pipeline sampling and learning pseudo code can be found in Appendix A.

### 3.1. Data Pre-Processing

Data pre-processing takes WiFi-CSI data in the $C^{\left(A\times K\times T\right)}$ format, where *A* denotes the transceiver link antenna and *K* sub-carrier. The reason for calling *A* the transceiver link antenna is, when indexing the transmitter and receiver dimensions, for the downlink or uplink sampled WiFi-CSI data, the data measured at the antenna quantifies the state of a specific transmitter/receiver link (we bastardize this in the term ‘transceiver’ in this case). During the experiments we considered two datasets: a Widar3 data subset and the SignFi dataset [4,6]. Considered pre-processing steps per dataset are not always similar, but share similar steps, i.e., (i) reconstructing original phase values across time and sub-carrier, (ii) removing phase Sampling Time Offsets (STOs) and Frequency Time Offsets (FTOs) across sub-carrier and transceiver link antenna, (iii) filtering static and high frequency noise, (iv) stacking complex number amplitude and phase values in the sub-carrier dimension, and (v) zero padding/truncating time and sub-carrier dimensions to the nearest power of 2 to allow more variance in feature extractor depth during hyperparameter tuning. Pre-processing methods were taken from the original dataset papers [4,6] and a research project that also considered the aforementioned datasets [9]. More in-depth pre-processing method explanations can be found in [4,6,9].

WiFi-CSI data in the Widar3 subset is initially phase-unwrapped one-dimensional-wise across the time and sub-carrier dimensions in subsequent order. When a subsequent phase value (measured in radians) has an absolute difference of at least π with regards to its predecessor, the subsequent phase value is offset by adding the term 2kπ, where symbol k denotes the amount of times 2π is added until the absolute difference is lower than π. Data are then reduced from a given number of transceiver link antenna’s in the *A* axis to 2 such that the first antenna matrix contains far greater amplitudes than the second antenna matrix [24]. Amplitude adjustments are subsequently made to both antenna matrices. These adjustments are explained in more detail in [25]. Reduction and amplitude adjustments are normally considered to explicitly adhere to the Doppler shift extraction requirements and to increase the Doppler shift amplitude in subsequent pre-processing steps. Even though we do not explicitly extract Doppler shift information during pre-processing, reduction helps reduce computational/memory complexity in the feature extraction pipeline block. In addition, the Doppler shift information is considered to be a good feature for exergame gesture classification based on previous literature [24]. Therefore, feature extraction is likely going to rely on this feature, among other features, in one of its hidden representations for correctly inferring gestures. We remove phase value STOs and FTOs with the conjugate multiplication. This is possible because the antennas are connected to the same radio frequency oscillator (1 transmit antenna—3 receive antenna link mapping). Afterwards, a low and high-pass Butterworth [26] filter were applied. The low and high-pass filters are in the 6th and 3rd order, respectively and use a critical frequency of 60/500 and 2/500 half-cycles per sample. Lastly, amplitude and phase values were stacked along the sub-carrier dimension and the resulting tensor is transposed and zero padded/truncated to form an overall input tensor $\in R^{\left(T\times K\cdot 2\times A\right)}$. Performed gestures cause a transmitted signal to diffract, reflect, absorb, etc., across time and space in unique patterns. Superpositioned signal components, making up a received signal, are noted at the receiver. Unique phase and amplitude signal differences encoded in the overall input tensor quantify these unique patterns caused by finger, hand, etc., movements measured relative to the environment.

WiFi-CSI data in the SignFi dataset is also initially phase-unwrapped one-dimensional-wise across the time and sub-carrier dimensions in subsequent order. We removed phase value STOs and FTOs by the offset term addition to be found in Equation (Equation 1). Symbol *n* denotes the dataset index, ∠h CSI phase, $f_{\delta }$ frequency spacing between subsequent sub-carriers, and ξ offset term matrix constructed via multi-parameter linear regression. Minimization objective for optimal regression parameters to construct the offset term matrix can be found in Equation (Equation 2). Symbols η, ω, and β denote learned linear regression parameters. Term 2π(a−1) denotes the relative antenna phase difference with regards to an initial antenna caused by physical distance. The dataset we used for linear regression is SignFi dataset, of which the first time instant of every WiFi-CSI phase sample is taken ($\in R^{N\times A\times K\times 0}$). More information regarding linear regression can be found in [4]. By employing linear regression, we assume the optimal regression parameters are able to properly represent hidden STOs and FTOs. Conjugate multiplication is not possible because transceiver links do not share the same radio frequency oscillator (3 transmit antenna—1 receive antenna link mapping). Subsequently, a third-order Finite Impulse Response (FIR) bandpass filter is applied [9]. The filter was applied twice across the time dimension, once forward and once in reverse. Cutoff frequencies were 2 and 80 Hz, respectively. The filter uses a hamming window. Lastly, amplitude and phase values were stacked along the sub-carrier dimension and the resulting tensor was transposed and zero padded to form an overall input tensor $\in R^{\left(T\times K\cdot 2\times A\right)}$.
(1)$$\begin{array}{c} \hat{∠h_{n,a,k,t}}=∠h_{n,a,k,t}+2\pi f_{\delta }\left(k-1\right)\xi_{a,k} \end{array}$$
(2)$$\begin{array}{c} \underset{\omega }{\mathrm{argmin}\;}\sum_{a,k}{\left(∠h_{a,k}+2\pi \left(a-1\right)\eta +2\pi f_{\delta }\left(k-1\right)\omega +\beta \right)}^{2} \end{array}$$

#### Mini-Batch Creation

The last pre-processing step involves creating mini-batches of view combinations. Prior to creating mini-batches, all pre-processed samples belonging to the same gesture and repetition under the same unique combination of domain factors, but belonging to different transceiver links, are depth stacked together along transceiver link antenna dimension *A*. In the pre-training phase, when considering domain classification, a single combination is always made under similar domain factors. Let us assume the mini-batch size of 10. The mini-batch then has 10 positive view combinations, of which views belonging to a specific combination run through the network in a series (denoted by black/blue view flows in Figure 1). The unsupervised contrastive loss considered during the experiments called decoupled NT-Xent creates both positive and negative view combinations of every possible view combination as part of one of its computation procedures (see Sim in Equation (Equation 3)).

While using the Widar3 data subset as an input dataset, positive view combinations were made by splitting the antenna transceiver link dimension *A* of size 12 (6 tranceiver links · 2 antennas) in two dimension groups of size 6. During the fine-tuning phase, to prevent antenna transceiver link dimension mismatches, predictions resulting from the dimension groups made in a series were averaged to produce a final prediction. In the case of using the SignFi dataset as input dataset, a positive view combination denotes pre-processed input samples for two repetitions of the same class, while a negative view combination denotes samples belonging to either different or similar classes. The reason for this decision is that the antenna transceiver link dimension *A* in the SignFi dataset is 3 (an uneven number requires information duplication in the antenna transceiver link dimension which increases input data information redundancy, potentially leading to suboptimal results). In the fine-tuning phase, input data were provided as is without creating view combinations. The view combination creation step for the SignFi dataset, when mapped over training subset indices, assumes the available knowledge regarding the training subset indices, where samples denoting a similar class are stored for a specific training subset index. Views in a combination are fed separately in a series into the feature extractor.

### 3.2. Feature Extraction

Feature extraction involves one input modality $x\in R^{\left(T\times K\cdot 2\times A\right)}$. Symbol *T* denotes time dimension, K·2 denotes amplitude/phase stacked sub-carrier dimension, and *A* denotes transceiver link antenna dimension. The input modality denotes a single view. As explained previously, views are fed in a series into the feature extractor. The feature extractor illustrated in Figure 1 maps the input view to a latent representation.

Feature extractor hyperparameter details can be found in Table 1. The normal convolution operator does not consider the bias addition. The normal convolution and average pooling operators apply padding to keep input and output dimensions the same (e.g., `same’ padding). Batch normalization is always applied to the operator output before the activation function is applied.

When looking at Table 1, it can be observed that the feature extractor, after an initial convolution operator, contains MobileV2 operators. The MobileV2 operator is a computational complexity optimized convolution operator. It first convolves separate kernels over every input channel individually. Subsequently, it applies a number of pixel-wise convolutions equal to the desired output channel size. Additional MobileV2 operator details can be found in [17]. We consider MobileV2 operators because we expect overall gesture recognition solutions to be run on application-specific integrated circuitry combined with the radio hardware required for collecting WiFi-CSI data. Therefore, gesture recognition solutions need to be highly computational and complexity optimized to infer in an amount of time reasonable for the considered application using gesture recognition. In addition, cross-validation experiments in WiFi-CSI domain shift research typically require many model training runs. A large number of highly optimized operators inside the feature extractor significantly reduce the total time required to finish the experiments. The reason for picking MobileV2 operators which do not involve a squeeze-and-excitation mechanism [27] is to prevent multiple domain shift mitigation techniques being under testing in an experiment. Squeeze-and-excitation is often considered a channel attention mechanism, i.e., a domain shift mitigation technique type. Involving this type would make it infeasible to correctly validate the hypothesis 1 mentioned in Section 1. Because MobileV2 operator strides are always 1 × 1, no shape mismatches occur when adding a skip connection. Therefore, we consider no linear input mapping.

Interesting to note is that the average pooling operators consider a dynamic pooling stride value in the time dimension that is dependent on the input time dimension size. Dynamic values can be considered because the average pooling operator has no learnable parameters. The dynamic size prevents I[0]/2, i.e., one of the last average pooling operator stride values, becoming too large and causing architecture to overfit due to information loss. The increased pooling aggressiveness in the time dimension when the Widar3 subset was used did not cause overfitting problems. In addition, the dynamic size helps keep a universal latent representation size irrespective of differences in the input time dimension size. Learnable feature extractor parameters were initialized randomly by drawing from a variance-scaled normal distribution with a scale factor of 2 and `fan_out’ mode.

Hyperparameters mentioned in Table 1 were obtained via automatic hyperparameter tuning. The hyperparameter tuning algorithm considered is called hyperband optimization [28]. Considered hyperparameter tuning algorithm settings include an epoch budget of 600 per iteration, 5 iterations, and a model discard proportion per iteration of 3. An end-to-end supervised training setting is considered involving pre-processing, train subset augmentation, feature extractor, and gesture classifier. Gesture classifier details and augmentation details can be found in Section 3.4 and Section 4.3, respectively. The best set of hyperparameters was picked based on the lowest observed cross-entropy loss between gesture labels and predictions based on the held out validation set. The tuning was performed once for the SignFi downlink lab data subset involving users 1–5. Data samples were split randomly via stratified sampling according to a 80%/20% train/test split. Train split is further split into 75%/25% train/validate split. We used a random seed 42 and no view combination creation.

### 3.3. Projection and Domain Discrimination

The pre-training phase maps view combinations to a new embedding space, i.e., producing intermediate representation projections ${\mathrm{proj}}_{i}\;\mathrm{for}\;i\in \left\{1,2\right\}$. For every view combination, the projections are stacked on top of each other with pattern $\left\{{\mathrm{proj}}_{1,1},{\mathrm{proj}}_{1,2},\cdots ,{\mathrm{proj}}_{Bsize,1},{\mathrm{proj}}_{Bsize,2}\right\}$ and eventual structure $\in R^{\left(2\cdot Bsize,P\right)}$ before being given to the loss function. From now on, we refer to this structure as `proj’. Symbol *P* is the projection vector size. The projection network is a MultiLayer Perceptron (MLP) with 3 feed-forward layers, which use 450, 300 and 150 neurons, respectively. The first two layers include a ReLU activation function. The last layer linearly maps its input to the projection space (does not consider an activation function). All layers do not consider bias addition. Learnable parameter initialization is based on random samples taken from a variance scaled uniform distribution with a scale factor of 1/3 and `fan_in’ mode.

Our pipeline, when considering standard cross-entropy domain classification, also predicts domains under which view combinations were measured. This means that it returns the discrete probability distribution ${\left\{P\left(s_{i}|x_{1},x_{2}\right)\right\}}_{i=1}^{n_{domains}}$. Symbol $s_{i}$ denotes a one-hot encoded domain label. Views within a specific view combination always have to be measured under similar domain factors in this case. Therefore, a negative view combination in the case of the SignFi dataset only involves views of different classes measured under similar domain factors. The discriminator is a MLP with 4 feed-forward layers. These layers use 600, 450, 300 and 150 neurons, respectively. The first three layers include a ReLU activation function. The last layer includes a SoftMax activation function. Bias addition and learnable parameter initialization settings are similar compared to the projection network.
(3)$$L=L_{u}-\beta L_{d}$$
$$\begin{array}{c} L_{u}=\frac{1}{2\cdot Bsize}\sum_{i=1}^{Bsize}\left[l\left(2i-1,2i\right)+l\left(2i,2i-1\right)\right]\\ l\left(x,y\right)=-\ln\frac{e^{Sim_{x,y}/\tau }}{\sum_{k=1}^{2\cdot Bsize}1\left[k\neq x,k\neq y\right]\cdot e^{Sim_{x,k}/\tau }}\\ Sim=\left[\begin{array}{ccc} \frac{{\mathrm{proj}}_{1}^{⊤}\cdot {\mathrm{proj}}_{1}}{∥{\mathrm{proj}}_{1}∥\cdot ∥{\mathrm{proj}}_{1}∥} & \cdots & \frac{{\mathrm{proj}}_{1}^{⊤}\cdot {\mathrm{proj}}_{2\cdot Bsize}}{∥{\mathrm{proj}}_{1}∥\cdot ∥{\mathrm{proj}}_{2\cdot Bsize}∥}\\ \cdots & \cdots & \cdots \\ \frac{{\mathrm{proj}}_{2\cdot Bsize}^{⊤}\cdot {\mathrm{proj}}_{1}}{∥{\mathrm{proj}}_{2\cdot Bsize}∥\cdot ∥{\mathrm{proj}}_{1}∥} & \cdots & \frac{{\mathrm{proj}}_{2\cdot Bsize}^{⊤}\cdot {\mathrm{proj}}_{2\cdot Bsize}}{∥{\mathrm{proj}}_{2\cdot Bsize}∥\cdot ∥{\mathrm{proj}}_{2\cdot Bsize}∥} \end{array}\right]\\ L_{d}=-\frac{1}{|\hat{S}|}\sum_{i=1}^{|\hat{S}|}\sum_{j\in c_{d}}s_{i,j}\cdot g\left({\hat{s}}_{i,j}\right)\\ g\left(x\right)=\left(x-1\right)-\frac{{\left(x-1\right)}^{2}}{2}+\frac{{\left(x-1\right)}^{3}}{3},\cdots ,-\frac{{\left(x-1\right)}^{100}}{100} \end{array}$$

When considering multi-label domain classification, our pipeline predicts multiple domains, under which view combinations were measured. This means that it returns Bernoulli logits [29,30] per class ${\left\{Logit\left(s_{i}|x_{1},x_{2}\right)\right\}}_{i=1}^{n_{domains}}$. Logits are created linearly (without activation function involvement after the last feed-forward layer) and refer to model output variables prior to being used in a sigmoid activation function to create Bernoulli distributions [30]. Symbol $s_{i}$ denotes a multi-label domain vector. This vector differs from a one-hot encoded label in that at least 1, instead of at most 1, positive class indications can be present. The domain discriminator MLP network in this case is almost similar in network hyperparameters with regards to the projection network, except for the number of neurons in the last feed-forward layer. This number is equivalent to the number of domains for which a prediction needs to be made when multi-label domain classification involves the view of addition and half the number of domains when multi-label domain classification involves the view of concatenation.

The projection network and domain discriminator outputs are pre-trained with an overall adversarial loss presented in Equation (Equation 3). Subtracting $L_{d}$ (domain loss) ensures that this loss component is maximized. Symbol β is a weight (hyperparameter) controlling the impact of the domain loss on the overall loss. The unsupervised contrastive loss $L_{u}$ is based on decoupled NT-Xent loss. It brings views of the same classes measured under the same domain factors (i.e., positive combinations) closer together in the intermediate representation space, while forcing other view combinations (i.e., negative combinations) further apart. This ensures that structural information shared between views (e.g., information related to a gesture) is retained while view-specific factors (e.g., noise at specific receiver due to a hardware component malfunction) are ignored [31]. Loss details are explained in [32]. For a concise algorithmic overview, one may refer to [19]. The reason we do not consider the normal version of this loss [33] is that it overfits at lower batch sizes much quicker when performing experiments [32]. Symbol τ is a hyperparameter that controls the level of the feature extractor output representation negative and positive combination separation.
$$\begin{array}{c} S^{mes}={\left[\begin{array}{ccc} s_{1}+s_{1} & \cdots & s_{1}+s_{2\cdot Bsize}\\ \cdots & \cdots & \cdots \\ s_{2\cdot Bsize}+s_{1} & \cdots & s_{2\cdot Bsize}+s_{2\cdot Bsize} \end{array}\right]}_{⊥0}^{⊤1}\\ {\hat{S}}^{mes}=\left[\begin{array}{ccc} \sigma ({\hat{s}}_{1}^{logit}+{\hat{s}}_{1}^{logit}) & \cdots & \sigma ({\hat{s}}_{1}^{logit}+{\hat{s}}_{2\cdot Bsize}^{logit})\\ \cdots & \cdots & \cdots \\ \sigma ({\hat{s}}_{2\cdot Bsize}^{logit}+{\hat{s}}_{1}^{logit}) & \cdots & \sigma ({\hat{s}}_{2\cdot Bsize}^{logit}+{\hat{s}}_{2\cdot Bsize}^{logit}) \end{array}\right]\\ l\left(x,y\right)=\frac{1}{N}\sum_{m=1}^{N}(S_{x,y,m}^{mes}\cdot g{\hat{S}}_{x,y,m}^{mes}+\left(1-S_{x,y,m}^{mes}\right)\cdot \\ g\left(1-{\hat{S}}_{x,y,m}^{mes}\right))\cdot \left(1-I_{x,y}^{2\cdot Bsize}\right)\\ g\left(x\right)=\left(x-1\right)-\frac{{\left(x-1\right)}^{2}}{2}+\frac{{\left(x-1\right)}^{3}}{3},\cdots ,-\frac{{\left(x-1\right)}^{100}}{100} \end{array}$$
(4)$$L_{d}=\frac{1}{{\left(2\cdot Bsize\right)}^{2}}\sum_{i,j}\left[\begin{array}{ccc} l(1,1) & \cdots & l(1,2\cdot Bsize)\\ \cdots & \cdots & \cdots \\ l(2\cdot Bsize,1) & \cdots & l(2\cdot Bsize,2\cdot Bsize) \end{array}\right]$$

Standard domain discrimination loss $L_{d}$ is based on a Taylor approximated version of [34] multi-class cross entropy between the predicted domain label value ${\hat{s}}_{i,j}$ (0–1) and true domain label value $s_{i,j}$ (0 or 1) [34]. Symbol $|\hat{S}|$, in addition to Bsize, denotes the mini-batch size and $c_{d}$ the set of domain labels. At weight levels β close to the hidden unknown ideal, the domain discrimination loss, when involving standard cross-entropy as the loss function, explodes during training due to the log function output [34], approximating inf when the difference between the predicted and true domain label becomes larger. The overall loss is very sensitive to this explosion (loss progress stalls already after small loss explosions at values ≧30). The 100th order Taylor approximation puts a bound on the domain discrimination loss value [0−5.17], thereby absolving loss explosion. Multi-label domain classification loss uses parts of the decoupled NT-Xent loss similarity matrix creation procedure to create both prediction and multi-label domain tensors $S^{mes}\mathrm{and}\;{\hat{S}}^{mes}$. Letters mes refer to the fact that $S^{mes}\mathrm{and}\;{\hat{S}}^{mes}$ are created via a meshgrid function. These tensors are subsequently used together with an element-wise Taylor approximated binary cross-entropy loss function. The idea behind the multi-label domain classification loss (see Equation (Equation 4)) is that it successfully acts as an augmenter for hard domain confusion situations where the domain classification loss does not increase due to providing domain classification loss for all view combinations (i.e., more samples) instead of only positive view combinations. Prior to being given to the multi-label domain classification loss function, $s\;\mathrm{and}\;{\hat{s}}^{logit}$ are stacked on top of each other according to patterns $\left\{s_{1},s_{1},\cdots ,s_{Bsize},s_{Bsize}\right\}$ and $\left\{{\hat{s}}_{1,1}^{logit},{\hat{s}}_{1,2}^{logit},\cdots ,{\hat{s}}_{Bsize,1}^{logit},{\hat{s}}_{Bsize,2}^{logit}\right\}$, respectively. Symbols $\sigma ,⊤,⊥,\;\mathrm{and}\;(1-I^{N})$ refer to the sigmoid activation function, value clipping procedures, and an identity matrix of size N of which the values are flipped, respectively.

We manually defined both the projection and domain discrimination network parameters. For doing so, we were inspired by the projection network parameters presented in [19] and discrimination network parameters presented in [35].

### 3.4. Gesture Classification

After pre-training, the feature extractor network parameters were transferred to a supervised down-stream classifier. The classifier contains a gesture classification network instead of projection networks and a domain discriminator. When fine-tuning the classifier, the feature extractor network parameters were frozen and only the gesture classifier parameters were fine-tuned. Based on the type of view combination creation, a set of gesture classes was predicted either as is or the predictions per transceiver link antenna dimension group made in series were averaged to produce a final set of predicted gesture classes. The gesture classifier, in the fine-tuning phase, considers a multi-class cross-entropy loss (see Equation (Equation 5)). Symbol $|\hat{Y}|$, in addition to Bsize, denotes the minibatch size, $c_{a}$ denotes the set of gesture labels, $y_{i,j}$ denotes the gesture class label value (0 or 1) and ${\hat{y}}_{i,j}$ denotes the predicted gesture class label value (0–1).
(5)$$\begin{array}{c} L_{a}=-\frac{1}{|\hat{Y}|}\sum_{i=1}^{|\hat{Y}|}\sum_{j\in c_{a}}y_{i,j}\ln{\hat{y}}_{i,j} \end{array}$$

The gesture classifier is a MLP with two feed-forward layers. The first layer contains 300 neurons and uses a ReLU activation function. The last layer uses a SoftMax activation function. The number of neurons is based on the number of classes in the data subset(s) considered per experiment (276 or 150 for SignFi and 6 for Widar3; see Section 4.1). Bias addition and parameter initialization layer settings are similar to the projection network. The gesture classifier hyperparameters were manually defined.

## 4. Experimental Setup

We tested the hypotheses presented in Section 1 by means of cross-validation experiments. Dataset descriptions, baseline pipelines against which proposed pipeline was tested, and an evaluation strategy description, are presented below.

### 4.1. Datasets

For our experiments, we used the SignFi [4] and Widar3 [6] datasets. The SignFi dataset consists of WiFi-CSI data of many everyday sign gestures. The domain factors present in the dataset include two different environments, i.e., a home and a lab environment, and five different human test subjects. The lab environment has a much greater surface area ($156\;m^{2}$ versus $15.6\;m^{2}$), contains many more desks and cabinets, and does not contain items such as a bed or closet in comparison to the home environment. The test subjects vary in age, height, and weight in ranges of 26–39 years, 168–180 cm, and 55–90 kg, respectively. WiFi-CSI frames inside the dataset are raw unprocessed tensors $\in C^{\left(3,30,200\right)}$. The considered data collection setup consists of 1 transmitter with 3 co-located transmit antennas and 1 receiver with 1 receive antenna.

The experiments with the SignFi dataset involve both user/environment domain factor leave-out cross validation. During environment leave-out cross validation, the downlink home and lab data subsets for user 5 were considered. The home data subset contains 2760 samples (276 sign gestures·10 repetitions) and lab data subset of 5520 samples (276 sign gestures·20 repetitions). During user leave-out cross validation, a downlink lab data subset for users 1–5 was considered. This data subset contains 7500 samples (150 sign gestures·5 users·10 repetitions).

From the original Widar3 dataset, we constructed a subset including all repetitions of the following 6/22 gestures: push&pull, sweep, clap, slide, draw-O (horizontal), and draw-zigzag (horizontal). Regarding domain factors, we used data from environment 1 (classroom), all torso locations and face orientations, and user ids 10, 12, 13, 14, 15, and 16. Users vary in their age, weight, and height in ranges of 22–28 years, 45–87 kg, and 155–186 cm, respectively. The total size of this subset dataset is 6gestures·5gesturerepetitions·5torsolocations·5faceorientations·6userids=4500samples. For every domain factor leave-out cross-validation experiment, the same Widar3 data subset was used. The considered data collection setup consists of 1 transmitter with 1 transmit antenna and 6 spatially distributed receivers, each with 3 co-located receiver antennas.

### 4.2. Domain-Label-Based Domain Shift Mitigation Techniques

We compared the performance of our proposed pipeline, i.e., standard (ADV-S) and multi-label domain classification (ADV-M1) in the pre-training phase, with several other pipelines. Some pipelines consider the use of domain labels in a slightly different way compared to the adversarial domain classification during pre-training. When considering the multi-label domain classification in the pre-training phase, we also consider a loss (see Equation (Equation 4)) version, in which ${\hat{S}}^{mes}$ is made up by $\sigma ({\hat{s}}_{*}^{logit}⊕{\hat{s}}_{*}^{logit})$ (ADV-M2). In this case, the feature vector size of ${\hat{s}}_{*}^{logit}$ is halved. The difference between our proposed pipeline and  [8] is illustrated in Figure 1 and is related to (i) the integration of the adversarial domain classification in the pre-training phase in our pipeline and (ii) the way view combinations are fed into the pipeline. The reason why not all techniques presented in this section are considered baselines is because they also utilize domain labels during unsupervised pre-training, albeit differently when compared to the domain classification.

An overview of the computation resources that we have used and average required wall-clock time per epoch per pipeline can be found in Table 2. Interestingly, the average wall-clock execution time per pipeline doubles for some of the pipelines, even though the pipelines are placed onto a machine with approximately 6x the resources in comparison to the other pipelines. This indicates that the execution time increase does not depend on the computation resources, but rather a choice in the structure of the pipeline. The DAN-B pipeline uses costly additional unbatching and shuffle functions (see Section 5.2). In the ALT-F pipeline, simultaneous sampling operations for a pre-train and fine-tune path (see Section 4.2.5) which have a programmatic pipeline dependency on each other for the loss computation are performed. The pipeline structure choices mentioned previously, in addition to experienced memory leaks (see Section 5.2), also lead to a large RAM size requirement. The reason for using GPUs with large GPU memory is because pipelines incorporating a form of unsupervised contrastive pre-training generally require larger batch sizes in comparison to end-to-end supervised learning in order to be trained effectively. At lower batch sizes, loss instability issues are typically encountered, albeit to a lesser degree when considering a decoupled contrastive loss (see Section 3.3). No computation resource and wall-clock time per epoch information for the ALT-B pipeline is available because no machine was available that could support the RAM requirements of this pipeline.

#### 4.2.1. End-to-End No Domain Shift Mitigation (STD)

This baseline only considers the pre-processing, feature extractor, and gesture classifier blocks presented in Figure 1a. Architecture hyperparameters similar to the ones presented in Section 3 were used. The entire pipeline was trained via standard loss minimization end-to-end in a supervised learning style. This means that no learnable pipeline parameters were frozen during the training process. We considered the same loss as the one presented in Equation (Equation 5). Input data were provided as is without creating view combinations for both the Widar3 and SignFi datasets. All other details were similar to the fine-tuning details presented in Section 3.

#### 4.2.2. Pre-Training No Domain Shift Mitigation (STD-P)

This baseline considers similar blocks inside the pipeline presented in Figure 1b, except for the domain discriminator block, which was not considered. Architecture hyperparameters, except for the domain discriminator hyperparameters, were similar to the ones presented in Section 3. The feature extractor and projection network are first pre-trained. Afterwards, the projection network is discarded and the feature extractor’s learnable parameters frozen (excluded from the fine-tune training process). Subsequently, a gesture classifier network is stacked on top of the feature extractor. Afterwards, during fine-tuning, the gesture classifier network’s learnable parameters are trained. During pre-training, view combinations were mapped to an embedding space and structured according to `proj’. The overall loss, in which `proj’ was used, only considers the decoupled NTXENT loss $L_{u}$ in Equation (Equation 3). All other pre-training details and fine-tuning details were similar to the ones presented in Section 3.

#### 4.2.3. Pre-Training Domain-Aware Filter NTXENT (DAN-F)

This pipeline uses the same blocks as the pipeline presented in Figure 1b, except for the domain discriminator block, which was not used. Architecture hyperparameters, except for the domain discriminator hyperparameters, were similar to the ones presented in Section 3. The feature extractor and projection network are first pre-trained. Afterwards, the projection network is discarded and the feature extractor’s learnable parameters frozen (excluded from the fine-tune training process). Subsequently, a gesture classifier network is stacked on top of the feature extractor. Afterwards, during fine-tuning, the gesture classifier network’s learnable parameters are trained. During pre-training, view combinations were mapped to an embedding space and structured according to `proj’. The overall loss, in which `proj’ was used, only considers a special version of the decoupled NTXENT loss $L_{u}$, called domain-aware NTXENT loss [36]. An overview of the changes made to the loss component l(x,y) can be found in Equation (Equation 6). All other loss aspects were similar to the ones presented in Equation (Equation 3). When considering the SignFi dataset, positive view combinations were only based on similar task labels. Therefore, combination views may or may not originate from a similar domain. All other pre-training details and fine-tuning details were similar to the ones presented in Section 3.
$$S^{sim}=\left[\begin{array}{ccc} s_{1}=s_{1} & \cdots & s_{1}=s_{2\cdot Bsize}\\ \cdots & \cdots & \cdots \\ s_{2\cdot Bsize}=s_{1} & \cdots & s_{2\cdot Bsize}=s_{2\cdot Bsize} \end{array}\right]$$
(6)$$l\left(x,y\right)=-\ln\frac{e^{Sim_{x,y}/\tau }}{\sum_{k=1}^{2\cdot Bsize}1\left[\begin{array}{c} k\neq x,\\ k\neq y,\\ S_{x,k}^{sim}=1 \end{array}\right]\cdot e^{Sim_{x,k}/\tau }}$$

Looking at Equation (Equation 6), it can be noted that, in addition to positive view combinations, any view combination, of which the input sample views were sampled in the context of a different domain, is also filtered. Researchers proposing this loss claim that the repelling nature of NTXENT’s denominator component, i.e., if views are sampled in the context of a different domain, leads to a domain discriminative feature space (the exact opposite of a domain-invariant feature space) [36]. In the context of domain-aware augmentation, this loss is theoretically ideal for achieving domain invariance, since positive combinations are always sampled in the context of a different domain and thereby serve as extra attraction stimulus in the loss’s numerator. A limitation of this is that this positive view combination situation cannot be replicated (positive combinations may originate from both different and similar domains) without introducing a batch sampling bias.

#### 4.2.4. Pre-Training Domain-Aware Batch NTXENT (DAN-B)

This pipeline considers similar blocks inside the pipeline presented in Figure 1b, except for the domain discriminator block, which was not considered. Architecture hyperparameters, except for the domain discriminator hyperparameters, were similar to the ones presented in Section 3. The feature extractor and projection network are first pre-trained. Afterwards, the projection network is discarded and the feature extractor’s learnable parameters frozen (excluded from the fine-tune training process). Subsequently, a gesture classifier network is stacked on top of the feature extractor. Afterwards, during fine-tuning, the gesture classifier network’s learnable parameters are trained. During pre-training, view combinations were mapped to an embedding space and structured according to `proj’. The overall loss, in which `proj’ was used only, considers the decoupled NTXENT loss $L_{u}$ depicted in Equation (Equation 3). In order to make the contrastive pre-training procedure domain-aware, thereby eventually leading to being able to create domain-invariant input representations, domain-aware batch NTXENT considers additional view mini-batch creation and Sim matrix creation steps as alternative to the view mini-batch creation steps explained in Section 3.1. These steps make sure that views in every positive combination present in a specific mini-batch originate from the same set of different domains. This means that in every training and validation subset, data should have been sampled under two different domains. Otherwise, the pre-training domain-aware batch NTXENT pipeline conceptually does not work. These steps also make sure all negative view combination similarities are computed based on similar domains. All other pre-training details and fine-tuning details were similar to the ones presented in Section 3.

Using the Widar3 dataset, views were first uncombined and shuffled. The reason for this is that when views originate from the same sample by transceiver link antenna dimension splitting, the class and domain for each view is always the same. Afterwards, we applied two grouping functions. The first grouping function groups views based on a similar task label. Therefore, views belonging to different domains may or may not end up inside a view combination. The second grouping function groups view combinations inside a mini-batch based on a key function $argmax\left(s_{1}\right)+n_{domains}*argmax\left(s_{2}\right)$ that determines whether the view combinations in the mini-batch consist of similar domain label combinations. Lastly, mini-batches of which all views, irrespective of the view combination, have a similar associated domain label, were filtered out of the pre-processing pipeline. This filter operation is required because the aforementioned key function does not explicitly prevent the creation of these mini-batches. The next created mini-batch either (depending on the structure of the underlying dataset, random operator seed, etc.) contains view combinations in which views all originate from the same combination of different domains or all views have an associated similar domain label. When using the Signfi dataset, positive view combinations were firstly made based on similar task labels only. Subsequently, a second grouping function groups view combinations inside a mini-batch based on the same key function when the Widar3 dataset was used. Lastly, mini-batches of which all views, irrespective of view combination, have a similar associated domain label, were filtered out of the pre-processing pipeline.

For both the Widar3 and Signfi datasets, in every square 4-element Sim matrix patch apart from diagonal patches, the patch diagonal elements were flipped to make sure all negative view combinations originate from a similar domain. An example class-domain combination set typically found in a Sim matrix when considering pre-training domain-aware batch NTXENT can be found in Figure 2. Even though this pipeline runs the risk of bias, resulting in reduced inference performance towards a specific way of creating batches, it does not suffer from negative view combinations being filtered. Bias may exist because an explicit switch is made from creating mini-batches in which views from positive view combinations originate from the same combination of different domains during pre-training to creating mini-batches completely at random during fine-tuning and inferencing. Therefore, its effectiveness against a pre-training domain-aware filter NTXENT and the adversarial domain classification pipelines needs to be evaluated.

#### 4.2.5. Semi-Supervised Alternating Flow Pipelines

Alternative versions of the DAN-F and DAN-B pipelines which are trained in a semi-supervised way are also considered. The pipelines trained in a semi-supervised way are compared against DAN-F and DAN-B to investigate the potential domain shift showing up in the output representations of the gesture classifier architecture layers during fine-tuning the DAN-F and DAN-B pipelines. This situation seems unlikely to occur when the feature extractor is already taught to prioritize extracting domain independent features during pre-training. However, we argue that any pipeline and training process (albeit end-to-end training, pre-training, fine-tuning, etc.) involving learnable neural network parameters that are not constrained during a training process via a domain shift mitigation technique should be put under test to rule out potential domain shift influences showing up during the inferencing stage. The pipelines are called ALTernating domain-aware Filter NTXENT (ALT-F) and ALTernating domain-aware Batch NTXENT (ALT-B). An architecture overview compatible to both pipelines is illustrated in Figure 3.

During training iterations within a specific epoch, view combinations originating from a large pre-training dataset are given to the projection network pathway. Subsequently, samples originating from a much smaller dataset normally considered for fine-tuning are given to the gesture classifier pathway. Gradient updates for the respective pathways during each training iteration happen in sequence based on separate decoupled NTXENT loss $L_{u}$ and multi-class cross-entropy loss (see Equation (Equation 5)). All other pipeline details are the same as the details presented in the respective DAN-F and DAN-B pipeline subsections.

By alternating the pre-training and fine-tuning processes at the same time, we argue that the potential performance drops on a held out test set, caused by an introduced domain shift during fine-tuning in the gesture classifier output representations, can be alleviated. Performance drop alleviation is caused by the domain mitigation constraint the pre-training pathway introduces when alternating pre-training and fine-tuning. In the case of no alternation and leaving the fine-tuning as a subsequent training process altogether, a domain shift may or may not show up due to no domain shift mitigation constraint being present.

### 4.3. Evaluation Strategy

All pipelines mentioned in Section 3 and Section 4.2 were trained based on loss minimization. This subsection elaborates on the training and loss minimization details used during the experiments and that are necessary for experiment reproduction. Initially, details that are applicable to all pipelines explained in Section 3 and Section 4.2 are discussed. Additional details and detail deviations compared to the details applicable to all pipelines for respective pipelines are discussed in separate pipeline subsections.

Dataset splitting prior to pre-training was based on leaving out specific domain factors. This means that if 5 factor types exist for a specific factor, 20% will be placed into the test subset and 80% in the training subset. The Widar3 training subset is further split into a 17% validation subset and 83% training subsubset. Splitting was based on randomly leaving out specific domains (i.e., unique combination of domain factors). For SignFi, validation/training subset splitting happens randomly based on stratified sampling with task labels and a 25/75% split due to the low number of domain factors being present in each dataset.

Pre-training was performed by means of the Stochastic Gradient Descent (SGD) optimizer with extra regularization by means of weight decay and momentum. Additional regularization by means of early stopping after loss convergence or indications of overfitting was not considered due to overall noisy pre-training loss progression. The reason why the ADAptive with Momentum (ADAM) optimizer was not considered during the pre-training phase is that the loss function during experimentation constantly converged at a high loss value. A likely reason for this premature convergence is the inability of the learning rate scaling mechanism and standard ADAM hyperparameters that normally take care of reducing loss value oscillation around local minima to allow the loss value to surpass local minima boundaries. Epochs, batch size, learning rate, weight decay, Nesterov momentum, NT-Xent temperature, and random seed to introduce pseudo-randomness in random operations’ hyperparameters were the same for all domain factor leave-out cross-validation experiments, i.e., 100, 32, 0.0005, 0.000001, 0.95, 0.1, and 42, respectively (when applicable). The pipelines were trained for the complete number of epochs. Whenever a loss value obtained with the validation subset improved upon the last best predecessor, a new model parameter checkpoint was created. The best checkpoint at the end was loaded for fine-tuning and testing.

Epochs, batch size, learning rate, NT-Xent temperature, and random seed hyperparameter values used during pre-training were determined based on manual hyperparameter tuning. Weight decay and Nesterov momentum values used during pre-training were determined based on automatic hyperparameter tuning with the Bayesian optimization tuning algorithm [37]. We considered 20 maximum trials during Bayesian optimization. In addition, training subset augmentation was considered during Bayesian optimization (see subsequent paragraphs in this subsection for information). The best set of hyperparameters was picked based on the lowest observed decoupled NT-Xent loss based on a held out validation subset. Tuning was performed based on one SignFi downlink lab data subset split procedure involving users 1–5. Data samples, for hyperparameter tuning, were firstly split randomly via stratified sampling based on task label according to a 80%/20% train/test split. Secondly, the training dataset was split into 75%/25% training/validation data subsets (with 42 as a random seed). Lastly, the test data subset was discarded (hyperparameter tuning did not involve a test subset inferencing phase).

The fine-tune training and validation subsets were sampled from the pre-train training and validation subsets directly. This means no training and validation subset data mixing can occur when switching to the fine-tune phase. Data mixing can occur when you sample the fine-tune training and validation subsets directly from the original data(sub)sets. The pre-train training and validation subsets are randomly divided based on stratified sampling with task labels. The pre-train and validation division ratios are 75%/25% and 50%/50%, respectively. Subsequently, the 25% division from the pre-taining training subset and second 50% division from the pre-train validation subset are used as fine-tune training and fine-tune validation subsets.

Fine-tuning happened with the ADAM optimizer. The learning rate and batch size throughout all fine-tuning phases were set to 0.0001 and 16, respectively. The learning rate and batch size values were determined based on manual hyperparameter value tuning. Epoch number and random seed were the same as those during the pre-training phase. Weight transferring happened by copying weights to the fine-tuning model and setting all layers inside the feature extractors (see Figure 1) to the inference phase (set flags such that gradient computation skips feature extractors). Fine-tuning also considers loading a best performing checkpoint prior to testing the model with the held-out test subset. The best performing checkpoint is picked based on the lowest observed fine-tune loss value at the end of a specific epoch.

All training subsets, during pre-training and fine-tuning in all domain factor leave-out cross-validation experiments, were augmented on-the-fly after sampling. Augmentation was applied to partly alleviate sub-optimal performance metric test set results caused by the small size of the considered input datasets (Widar3 subset dataset and SignFi dataset). We considered a limited number of augmentations such as no augmentation, Gaussian noise overlay, time permutation (overall input tensor was broken up into time windows, and time windows are shuffled and concatenated into original overall input tensor shape), and random sample mixing with the considered input sample. We did not perform multiple augmentations in sequence. Every augmentation has a 25% chance to be picked (augmentation method sampling is based on picking a number between 1 and 4 from a uniform distribution with seed 42). The augmentations were directly taken from research studies involving Doppler Frequency Spectogram (DFS) and three-dimensional acceleration signal augmentation conducted by Zhang et al. and Um et al. [38,39]. Additional augmentations that were tested for potential performance improvements on a leave-out randomly in-domain sampled SignFi user validation subset with the STD pipeline include time, amplitude value, and phase value warping (overall input tensor is multiplied in the respective dimension with a generated highly non-linear curve), and amplitude and phase value scaling (overall input tensor is multiplied, or phase rotated, in the respective dimension with a constant number). None of these augmentations introduced noteworthy performance improvements.

After fine-tuning, classification performance metric results were obtained with the specifically sampled test data subset. Performance metrics considered include accuracy (A), precision (P), recall (R), F1 score (F), and Kappa score (CK).

#### 4.3.1. End-to-End No Domain Shift Mitigation (STD)

The STD pipeline is trained end-to-end in a supervised way. It therefore does not consider separate pre-train and fine-tune phases. The STD pipeline uses similar data sampling procedures as the earlier discussed pre-training phase to obtain training/validation/test subsets. This is possible due to the Independent and Identically Distributed (IID) nature of the considered datasets with regards to both task and domain labels. The considered optimizer and hyperparameters, including hyperparameter values, were the same as for the earlier discussed fine-tuning phase (see details applicable to all pipelines). End-to-end training in a supervised way does not consider the view combination creation during pre-processing.

#### 4.3.2. Pre-Training Single- and Multi-Label Domain Classification (ADV-S, ADV-M1, and ADV-M2)

The pre-train loss uses a domain loss weight (see Equation (Equation 3)). The considered domain loss weight was set to 0.1 for the Widar3 dataset and 0.01 for the SignFi dataset. The forementioned domain loss weight values were determined based on manual hyperparameter tuning. No domain loss weight distinction is made between single-label and multi-label domain classification. We consider manual separate domain loss weight hyperparameter values per dataset to be an inference performance generalization limitation of the ADV-S, ADV-M1, and ADV-M2 pipelines. When the pipelines are re-trained and subsequently used for inferencing in a task setting comparable to the one encoded in the Widar3 and SignFi datasets, there is a chance the domain loss weight has to be tuned again due to observed inference performance drops caused by suboptimal domain independent feature extraction. Ideally, the weight can remain static across datasets or can be adjusted automatically via a domain loss weight alteration algorithm in a so-called training warm-up phase prior to actual pipeline training. Unfortunately, no suitable domain loss weight alteration algorithm was found.

#### 4.3.3. Semi-Supervised Alternating Flow Pipelines (ALT-F and ALT-B)

The ALT-F and ALT-B pipelines, during training iterations within a specific epoch, are first given mini-batches (containing view combinations) originating from pre-training training and validation data subsets. The mini-batches are used in the projection network pathway. Subsequently, mini-batches originating from fine-tuning training and validation data subsets are given to the pipelines. The mini-batches are used in the gesture classifier pathway. Gradient updates for the respective pathways during each training iteration happen in sequence.

Both pre-training and fine-tuning training and validation subsets were created prior to the training phase. Sampling settings were similar to the pre-training and fine-tuning subset sampling settings applicable to all pipelines explained earlier. The fine-tune training and validation subsets do not consider view combination creation during pre-processing. Mini-batches sampled from the fine-tune training and validation subsets consider a batch size of 32 similar to the batch size value considered in pre-training training and validation subset mini-batches. Training iteration steps 1 and 2 (see Figure 3) use different optimizers. These optimizers, including their respective hyperparameters, are comparable to the pre-training and fine-tuning optimizers and associated hyperparameters applicable to all pipelines explained earlier. Loading a best performing checkpoint prior to testing the model with the held-out test subset is based on an overall summation between the loss obtained at the projection network and gesture classifier pathways at the subsequent training iteration steps. When considering the ALT-F and ALT-B pipelines, performance metrics are only computed for the gesture classifier pathway.

## 5. Experimental Results

In this section, we present and discuss our experimental results.

### 5.1. Domain Classification Integration Results

Figure 4 depicts the in-domain and leave-out one domain factor cross-validation results of the Widar3 data subset and stacked CSI amplitude and phase values as the input type. At a first glance, one may notice, in Figure 4a, the overall performance discrepancy between the STD and other pipelines in in-domain cross-validation experiments. Since view creation is based on the transceiver link dimension splitting, a reduction in overall input data volume for the other pipelines is not a likely culprit. Prior unsupervised pre-training research, when considering vision-based input datasets, has shown that model complexity and data volume requirements for unsupervised pre-training are much higher than for end-to-end supervised training [19]. Looking at the user factor leave-out cross-validation results in Figure 4b, a substantial increase in performance metric variance and mean P metric values over other metric values for all pipelines can be noted. After inspecting individual user leave-out cross-validation split results, a substantial mean A, R, F, and CK metric value drop (approx. 30% with regards to other split results) for one of the splits was noted. The mean P metric value drop for this split was less pronounced (approx. 15% with regards to other split results). No user factor data volume imbalances were found in the considered Widar3 data subset. One potential explanation for this situation is that a hidden subfactor, which is significantly different for the user being left out from the train and validation subsets, negatively influences the performance results. An example may be the pattern across time with which a specific gesture was performed. A similar, albeit less pronounced, performance drop for one of the user splits was noted in the Widar3 dataset research paper [6]. The difference in the performance drop can be attributed to dissimmilar pipeline and hyperparameter tuning choices that were made in our cross-validation experiments.

The model complexity and data volume requirement advantage of the STD pipeline, when leaving any of the available domain factors out, is influenced by a domain shift. As expected, when integrating domain classification in an unsupervised pre-training setting, and not leaving out a specific domain factor, no noteworthy performance differences are noted (see Figure 4a). Interestingly, the leave-out one domain factor experiment settings also do not show any noteworthy performance improvements for ADV-S, ADV-M1, and ADV-M2 over STD-P (see Figure 4b), except for the position (ADV-M1 has an average 1.2% mean value improvement across all metrics with a 0.5% variance increase) and orientation (ADV-M1 has an average 0.4% mean value improvement across all metrics with a 1.0% variance decrease) leave-out experiments (see Figure 4c,d). A potential culprit that explains this situation is the conflicting relationship between (i) confusing the feature extractor with a domain classifier by maximizing the domain classification loss and (ii) the decoupled NTXENT loss forcing distance between negative view combinations of which views were sampled from arbitrary domains. When a large proportion of arbitrary negative view combinations were sampled from different domains, the model is overstimulated to push views sampled from different domains further apart, thereby causing the feature extractor’s latent feature space to lead more towards a domain discriminative space. When this is not the case, maximizing domain classification loss and forcing a negative view combination distance form a harmonious relationship. Therefore, different hidden proportion ratios across sampled mini-batches result in an undeterministic domain shift reduction result for ADV-S, ADV-M1, and ADV-M2 in an unsupervised pre-training setting in one domain factor leave-out cross-validation experiments. The culprit and aforementioned proportion ratio situation are further discussed in Section 5.2.

Additionally, we present results of a discrepancy test between the A values measured on (i) the validation subset during fine-tuning/end-to-end training across time and (ii) on the test subset once the best performing pipeline train checkpoint was restored. The validation subset, as explained in Section 4.3, was sampled randomly in-domain, while the test subset was sampled out-domain based on leaving-out the first orientation domain factor during experiments. Discrepancy test results can be found in Figure 5. As expected, for both STD and STD-P pipelines, a value discrepancy between the validation and test subsets increases across time. Interestingly, the increase for ADV-S and ADV-M2 is less pronounced and seems to stagnate after a small number of epochs. For the ADV-M1 pipeline, a downward discrepancy trend can be noted during later epochs. These results are in line with the overall cross-validation results and imply that ADV-M1 has a better domain shift influence mitigation capability compared to ADV-S and ADV-M2.

Figure 6 depicts the leave-out one domain factor cross-validation results for the SignFi datasets and stacked CSI amplitude and phase values as the input type. The STD pipeline results (see Figure 6a) confirm the model complexity and data volume requirement observations that were made for the Widar3 subset. However, one may notice that there is more to it. In Section 3.1, it was explained that, due to the low and uneven number of transceiver links per input sample, we decided to aggregate samples in view combinations belonging to the same task and sampled under a similar domain in case of the STD-P, ADV-S, ADV-M1, and ADV-M2 pipelines. Therefore, in addition to the model complexity and data volume requirements, the SignFi datasets for these pipelines are effectively halved in terms of volume during pre-training. The decision to aggregate samples, and model complexity and data volume requirements, combined, have resulted in model overfitting. The feature extractor of this model was then used during fine-tuning. The performance discrepancy results depicted in Figure 7 support this overfitting claim. Comparing validation subset results between STD and STD-P, an approximate 20% A value decrease can be noted. Interestingly, a similar discrepancy increase between the validation and test subsets is noted for the STD-P, ADV-S, ADV-M1, and ADV-M2 pipelines without noticeable differences across epochs. We attribute the similar discrepancy increase to the overfitting during pre-training, effectively making the domain classification techniques incapable of reducing domain shift. We attribute the higher overall discrepancy between the validation and test subsets for the STD-P, ADV-S, ADV-M1, and ADV-M2 pipelines, when compared to the STD pipeline and the fine-tuning, as well as in combination with the overfitted feature extractor, leading to more domain shifts.

Typical approaches to prevent reducing the volume by half include (i) introducing an augmented version of an input sample within a positive view combination [19], (ii) prioritizing an even number of transceiver links/antennas within a transceiver link such that a split can be made within these dimensions to create a positive view combination [8], and (iii) sampling input sample subsections across time [40]. In Section 4.3; it is explained what augmentations we applied to increase variance among input samples. Other noteworthy augmentation methods did not introduce performance increases. The upscaling as augmentation may introduce unwanted redundancy (see Section 3.1) and downscaling can only happen under the assumption that input sample multipath components between antennas are similar [24], which is not applicable to the SignFi dataset.

Looking at the user factor leave-out cross-validation results (see Figure 6a), a substantial increase in performance metric variance and mean P values over other metric values for all pipelines can also be noted. However, when inspecting the individual split results, compared with the Widar3 results, a large variance among A, R, F, and CK values across all splits was noted. The P values are substantially higher for all splits with substantial variance, except for one of the splits which shows a significant drop with regards to other splits (approx. 30%). We consider higher P values, in addition to the lower R values, to be a sign that many true positive gesture classes were missed. However, the ones that were not missed have a higher likelihood to not be false positives. This is a typical indication that class distributions in the latent space are very narrow. This can be directly related to the low sample volume per gesture class in the SignFi datasets (i.e., 40 per class for the user dataset in comparison to 750 per class for the Widar3 subset). This increases the chance for more class distribution outliers with narrow class distributions overall. In this case, we recommend using n-shot learning (see Section 2.1), which performs better than unsupervised representation learning.

Looking at the environment factor leave-out cross-validation results (see Figure 6b), abysmal results in comparison to the user results can be noted. We attribute these results to two factors. (i) Not having retuned the hyperparameters for all pipelines on the environment SignFi dataset prior to conducting the environment factor leave-out cross-validation experiments. In the environment SignFi dataset, the task changes from recognizing 150 different gesture classes to recognizing 276 gesture classes. (ii) Even lower numbers of sample volume per gesture class (10/20 per class depending on which environment factor is being left out in comparison to 750 per class for the Widar3 subset). The class number change and drop in data volume per class significantly increases the chance of requiring hyperparameter retuning and data volume complexity and narrow class distribution risk when class decision is based on class distributions rather than sample distance comparison. Therefore, in this case we also recommend using n-shot learning over unsupervised representation learning.

When looking at the Widar3 random and domain factor leave-one-out cross-validation results presented in Figure 4, mean metric values between 70% and 85% can be observed for all baseline pipelines (STD and STD-P). The SignFi leave user 3 out discrepancy test results presented in Figure 7 show mean metric values between 65% and 75% on a randomly sampled left-out validation dataset with the aforementioned baselines when considering user domain factor leave-out cross-validation for all baseline pipelines. We attribute the baseline pipeline mean metric values to an input sample type that generalizes, in terms of overall solution performance, to both aforementioned datasets. If the input sample type was not generalizable, the mean metric values measured on the SignFi randomly sampled left-out validation dataset should have approached the mean metric values indicating near random guessing measured on the left-out test data subset for the STD-P pipeline.

Overall, the cross-validation results obtained with the SignFi datasets and stacked CSI amplitude and phase values as input types unfortunately show a low capability of mitigating the domain shift problem. Therefore, we do not consider the SignFi dataset further in our analysis.

### 5.2. Domain Classification and Domain-Aware NTXENT Results

We suggested the existence of a conflicting relationship between (i) confusing the feature extractor with a domain classifier by maximizing the domain classification loss and (ii) the decoupled NTXENT loss forcing distance between domain arbitrarily sampled negative view combinations. Figure 8 depicts the leave-out domain factor cross-validation results, in which the ADV-S, ADV-M1, and ADV-M2 pipelines are compared against domain-aware NTXENT pipelines (i.e., DAN-F and DAN-B). Authors of DAN-F [36] indicated that by filtering negative view combinations, distance cannot be forced anymore between similar task labeled views originating from different domains.

At first glance, Figure 8 seems to suggest the occurrence of the polar opposite (more distance is forced between similar task labeled views originating from different domains). The results overall show a drop between DAN-F and the ADV pipelines of approximately 20–25% depending on which domain factor is being left out. However, upon further inspection of the discrepancy results in Figure 9, it can be noted that the A values measured on the leave-out validation subset during fine-tuning slowly converges towards the eventual measured A values on the leave-out out-domain test subset without surpassing it. Interestingly, the convergence speed is much slower when compared to the ADV-S, ADV-M1, and ADV-M2 pipelines. We believe the discrepancy results indicate that DAN-F is able to mitigate a domain shift by filtering negative view combinations of which views were sampled from different domains. Since ADV-S, ADV-M1, and ADV-M2 still show a discrepancy, they do so suboptimally in comparison to DAN-F. This confirms the existence of the earlier suggested relationship.

However, something else is hampering the overall performance effectiveness of the DAN-F pipeline, as illustrated by the cross-validation performance result drop and slow convergence speed. We believe the reduction in overall performance is caused by the reduction in the number of negative view combinations eligible for loss computation further down the pipeline. Let us assume that the overall NTXENT loss is made up of m/n computed positive/negative view combination similarities. During training, *m*’s summation is minimized, while *n*’s summation is maximized. When many *n*’s are filtered, the loss value arbitrarily increases for a specific mini-batch. This teaches the pipeline that the current direction step in the loss space taken is bad irrespective of whether the step leads to better achieved performance, as indicated by a yes or no. This directly influences the potential positive benefit of reducing the distance between views sampled from a different domain. The eventual performance convergence can be explained by the fact that many *n*’s are not always filtered in every training iteration. As a result, fine-tuning can still use suboptimal structural information in the feature extractor. An inspection into the DAN-F pre-train NTXENT loss for the first orientation leave-out split has confirmed this view (the loss keeps alternating at values between −4 and −5 even in later epochs). Therefore, we recommend finding methods to reduce the number of *n* similarities being made ineligible for the eventual loss computation.

The DAN-B pipeline was assumed to have the ability to reduce the number of *n* similarities being made ineligible at the cost of a potential increased model bias towards constructing mini-batches based on positive view combinations, in which views were sampled from the same combination of different domains. The results in Figure 8, however, show an even worse overall performance drop with regards to the ADV-S, ADV-M1, and ADV-M2 pipelines. In addition, the performance is approaching the performance of a random guessing system (approximately 30%). In addition, the discrepancy results in Figure 9 suggest that no domain shift reduction has taken place at all throughout the epochs. Therefore, we can conclude that DAN-B is doing the polar opposite to what it was intended to do. However, upon closer inspection of the NTXENT pre-training loss and memory resource usage of this pipeline during pre-training (see Figure 10), a static high NTXENT loss across epochs and memory fragmentation/leakage across time were observed. The loss reaction is different from the observed loss on the SignFi dataset when running user leave-out cross-validation user split 2, which indicates overfitting. Therefore, we consider the DAN-B results to have originated from a randomly initialized feature extractor and gesture classifier optimized during fine-tuning. This explains results slightly above random guessing with an observable domain shift between validation and test subset. The effectiveness of DAN-B should be re-evaluated in future research once the static high NTXENT loss culprit has been found. The main differences between DAN-F and DAN-B relates to the pre-processing phase. In the DAN-B pre-processing pipeline, additional unbatching and shuffle functions are used to make sure views within a combination do not originate from the same domain as a result of combination creation by means of transceiver link antenna dimension splitting. Therefore, the culprit likely originates from technical faults within these functions or the interaction with aforementioned functions with other pre-processing functions.

### 5.3. Domain-Aware NTXENT and Alternating Flow Results

The leave-out cross-validation results for the comparison between the DAN-F and ALT-F pipelines can be found in Figure 11. The DAN-B and ALT-B pipelines have not been included due to the technical pre-training loss and computation resource difficulties (additional information is provided in Section 5.2). In addition, for the ALT-B pipeline to work, four, instead of two, simultaneous pre-processing pipelines would have to be loaded into RAM. This would theoretically double the witnessed computation resource results depicted in Figure 10c. No compute infrastructure was available to support these resource computation requirements at the time of experimentation.

Looking at the cross-validation results, an overall performance increase of approximately 10% from the DAN-F to the ALT-F pipeline can be observed. A potential underlying reason may be, even though the current direction step of the decoupled NTXENT loss is considered arbitrarily bad, the resulting effect of it is subsequently reduced by training the classification pathway in training iteration t+1. By looking at the discrepancy results in Figure 12, it can be noted that the discrepancy of the ALT-F pipeline is very noisy across epochs with large sawtooth jumps. This indicates that the classification and contrasting pathways influence each other. The discrepancy results also indicate that if a classification pathway is not explicitly constraint via a domain shift mitigation technique (e.g., constraint via an overall loss relationship used in one training iteration step), while being alternated with a contrastive pathway which takes into account domain awareness by means of negative view combination similarity filtering, domain shift is reduced at arbitrary epochs.

### 5.4. Domain Classification Integration Comparison against State-of-the-Art Pipelines

Lastly, we consider a comparison of the aforementioned pipelines with a state-of-the-art solution, i.e., WiGRUNT [22]. We select WiGRUNT for the comparison because its published results were obtained based on the same Widar3 subset we used for our experiments. No additional state of the art domain shift mitigation pipeline was found that considers the exact same SignFi dataset and Widar3 subset.

The WiGRUNT pipeline integrates two attention sub-networks into the neural network portion of the pipeline to reduce a domain shift. Attention refers to putting more cognitive attention towards parts of an input sample that matter for the learning task at hand. The first sub-network learns a function to produce an input mask based on the input sample and subsequently combines it with the input sample to prioritize input sample sections. The second sub-network learns a function to produce a mask based on the latent feature space. When combined with features resulting from the aforementioned space, it prioritizes feature relationships arbitrarily across time.

Figure 13 presents similar leave-out cross-validation results as presented in Section 5.1, but now compared against mean A values reported by the WiGRUNT paper. The WiGRUNT pipeline outperforms all pipelines by approximately 17%, 16%, and 12% for in-domain sampling, leaving out a position, and leaving out an orientation factor. This raises the question: why does this happen? Firstly, WiGRUNT considers phase heatmaps as the input type. This means amplitude information is not taken into account. Therefore, when considering the Widar3 subset, amplitude information could be redundant information causing lower overall performance. Secondly, all pipelines presented in Section 3 and Section 4.2 use a feature extractor which is based on MobileNetV2 blocks. WiGRUNT uses residual blocks inside a ResNet18 architecture. MobileNetV2 blocks are computation resource complexity optimized blocks, which come at the cost of reduced performance for at least a limited number of epochs (as is the case during experimentation). Lastly, WiGRUNT is trained in end-to-end supervised from scratch while all other pipelines, except for the STD pipeline, are first pre-trained and subsequently fine-tuned. Pre-training and fine-tuning has higher data volume and model complexity requirements in comparison to end-to-end supervised training [19].

### 5.5. Hypothesis Validation

In Section 1, a few hypotheses were drawn.

**Hypothesis** **1.**
*The influence of domain factors can be mitigated in a downstream task classifier by introducing adversarial domain classification in a deep neural network used in the unsupervised pre-training phase.*


This hypothesis was validated. When the NTXENT loss, and domain classification loss, form a harmonious relationship, the domain shift can definitely be mitigated. This is illustrated by experiment results with the Widar3 dataset, indicating that the ADV-M1 pipeline reduces a domain shift in position (1.2% mean metric improvement and 0.5% variance increase) and orientation (0.4% mean metric improvement and 1.0% variance decrease) in domain factor leave-out cross-validation experiments when being compared against the STD-P pipeline.

**Hypothesis** **2.**
*In situations where the downstream task classifier performance does not improve, or marginally improves, due to the multi-class and single label domain classifier used during pre-training failing to reduce the negative performance impact of domain factors, a multi-class multi-label domain classifier can be introduced instead. The multi-class multi-label domain classifier, during pre-training, does cause the eventual downstream task classifier performance to improve because the negative performance impact of domain factors is reduced. The multi-class multi-label domain classifier exploits the positive and negative view combination matrix created in the NT-Xent loss function during pre-training.*


This hypothesis was also validated. Where the ADV-S and ADV-M2 pipelines do not show noteworthy performance improvements over the STD-P pipeline during domain factor leave-out cross-validation experiments, the ADV-M1 pipeline does (see results mentioned in prior paragraph).

## 6. Conclusions and Future Work

In this paper, we proposed the integration of an adversarial domain classifier in the pre-training phase. The unsupervised representation learning pipeline that was considered during the experimentation uses view combination distance comparison during pre-training. Based on this pipeline, a few hypotheses were drawn in Section 1. As explained in Section 5.5, we validated the hypotheses.

In addition to hypothesis validation, a few noteworthy observations were made during pipeline experimentation. The considered input sample type generalizes, in terms of the overall solution performance, to both considered Widar3 and SignFi benchmark datasets. It should be noted that all other observations were noted based on the Widar3 subset. Observations, for the purpose of generalization, should be re-evaluated with (i) datasets focusing on domain shift reduction which becomes available in the future and (ii) view creation procedures which do not negatively impact pre-training data volume together with the SignFi datasets. When the NTXENT loss, and domain classification loss, do not form a harmonious relationship, the domain shift reduction capability of the domain classification pipelines (ADV-S, ADV-M1, and ADV-M2) is significantly reduced. More discord is created between the NTXENT loss and domain classification loss when a majority of negative view combinations used in the NTXENT loss contain views originating from different domains. In Section 3.3, we indicated that the idea of multi-class and multi-label domain classification is that it acts as a sample augmenter. Experiment results have shown that this is not entirely the case. Augmentation, or less influence of the negative relation with the NTXENT loss, could both be the reason that the ADV-M1 pipeline performs better in comparison to the ADV-S pipeline.

A way to overcome a majority of negative view combinations used in the NTXENT loss containing views originating from different domains is filtering negative view combinations of which views originate from different domains. Filtering can be ingrained into a special domain aware version of the NTXENT loss. However, the domain-aware contrastive loss based on filtering negative view combinations suffers from arbitrary negative loss direction indications. Structuring mini-batches such that positive views in every combination originate from different domains was suggested as a way to overcome arbitrary loss direction indications. However, due to technical limitations, the DAN-B pipeline, which uses the aforementioned mini-batch structure technique, could not be properly tested.

DAN-F and DAN-B pipeline variations, called ALT-F and ALT-B, which alternate contrastive learning with task learning in a single training iteration, were also introduced. The DAN-F and DAN-B pipelines were compared against these variations to test whether the variations mitigate a domain shift which shows up in the later classification sub-network feature representation spaces. The experiments showed that when a classification pathway is not explicitly constrained via a domain shift mitigation technique, while being alternated with a constrastive pathway which takes into account domain awareness by means of negative view combination filtering, the domain shift is reduced at arbitrary epochs.

For the future research, the highest priority should be put on envisioning domain-aware unsupervised contrastive learning pipelines, which limit view combination distance maximization between views originating from different domains. Examples when considering vision data include negative combination filtering and domain augmentation [36], and completely generating negative combinations [41]. In addition, batch structuring as suggested in this research paper is another example. Important to note is a set of requirements of these pipelines. The pipelines should (i) work at lower batch sizes without introducing arbitrary loss deviations, (ii) be commutable in a single graph sweep (as is the case with standard decoupled NTXENT loss) without introducing memory fetching operations, which introduce additional time complexity, (iii) limit the number of simultaneous pre-processing pipelines leading to technical memory complexity difficulties, and (iv) generalize to hidden domains. Another research direction related to the introduction of adversarial domain classification as a domain shift reduction boosting mechanism can be explored. Additionally, the result reproduceablity needs to receive high attention. This will allow being able to prove pipeline generalization. Finally, transfer learning may be a promising direction to follow as it makes the underlying domain shift effect stronger due to the hard domain factor separations between datasets and an additional domain factor, i.e., the considered data collection setup.

## Figures and Tables

**Figure 1 sensors-23-09233-f001:**
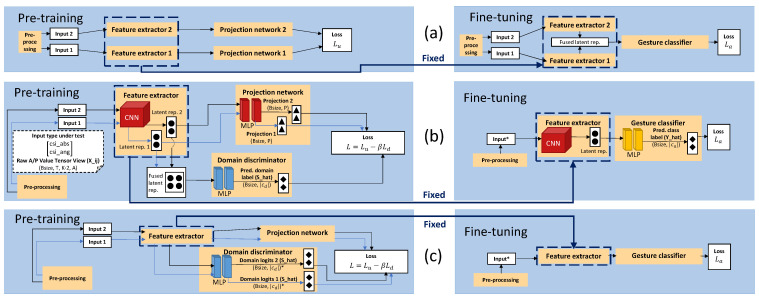
(**a**) Standard unsupervised contrastive learning approach of [8]. We show this approach to clearly illustrate the differences compared to our proposed unsupervised contrastive learning approach. (**b**,**c**) Architecture of our proposed unsupervised contrastive learning approach with integrated adversarial domain classification in the pre-training phase. (**b**) Domain classification. (**c**) Multi-label domain classification. Black/blue arrows denote input view flows that are fed in series into the architecture. (Bsize, $|c_{d}|$)*: if logit view combinations are concatenated during multi-label domain classification, $|c_{d}|$ is halved. Input *: A mismatches between pre-training and fine-tuning are handled during fine-tuning by duplicating A or producing average label prediction over groups of A. Approach sampling and learning pseudo code can be found in Appendix A.

**Figure 2 sensors-23-09233-f002:**
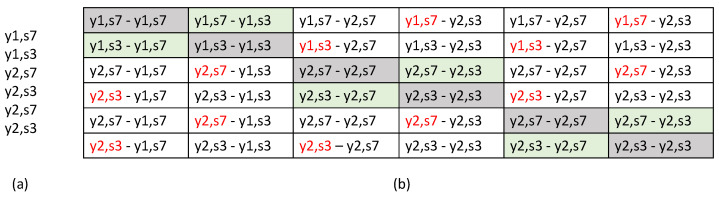
Class-domain combination set of a mini-batch, in which views within all positive view combinations originate from the same combination of different domains. (**a**) When view embeddings, class labels, and domain labels have been stacked according to pattern explained in Section 3.3 and (**b**) when Sim matrix has been created prior to the patch diagonal element flip procedure. Symbol *y* denotes class label and *s* domain label. Subsequent number refers to argmax location in label vector. Gray cell denotes view combination with itself. In (**b**), green cell denotes positive view combination. All other cells denote negative view combination. Red-colored diagonal elements in 4-element square patches not belonging to diagonal patches, when flipped, make sure all negative view combinations originate from a similar domain.

**Figure 3 sensors-23-09233-f003:**
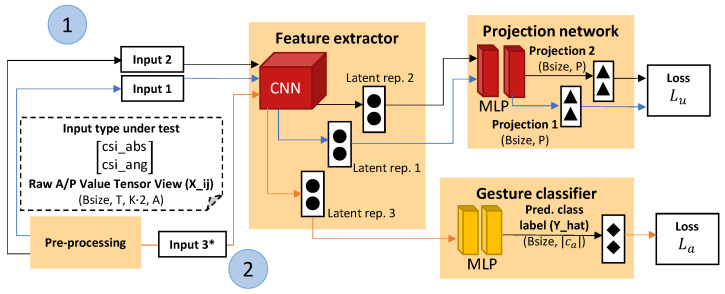
Architecture of the semi-supervised alternating flow pipeline. Blue-circled numbers indicate subsequent steps within a specific training iteration. Black/blue arrows denote input view flows that are fed in series into the architecture. Input 3*: A mismatches between steps 1 and 2 caused by having a feature extractor input channel equal to A of a specific input view are handled in step 2 by duplicating A or producing average label prediction over groups of A.

**Figure 4 sensors-23-09233-f004:**
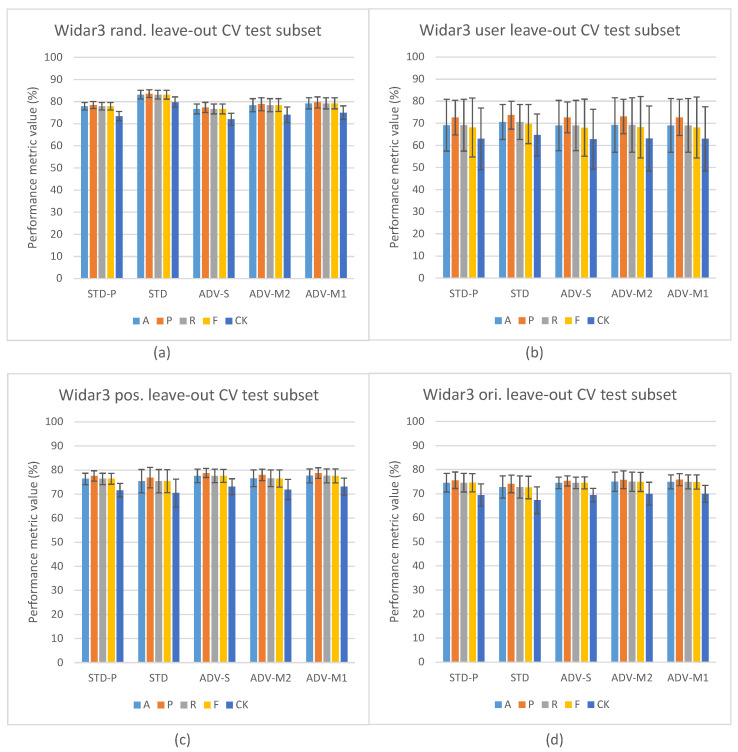
Widar3 domain classification (**a**) in-domain and (**b**) user, (**c**) position, and (**d**) orientation leave-out one domain factor Cross-Validation (CV) results.

**Figure 5 sensors-23-09233-f005:**
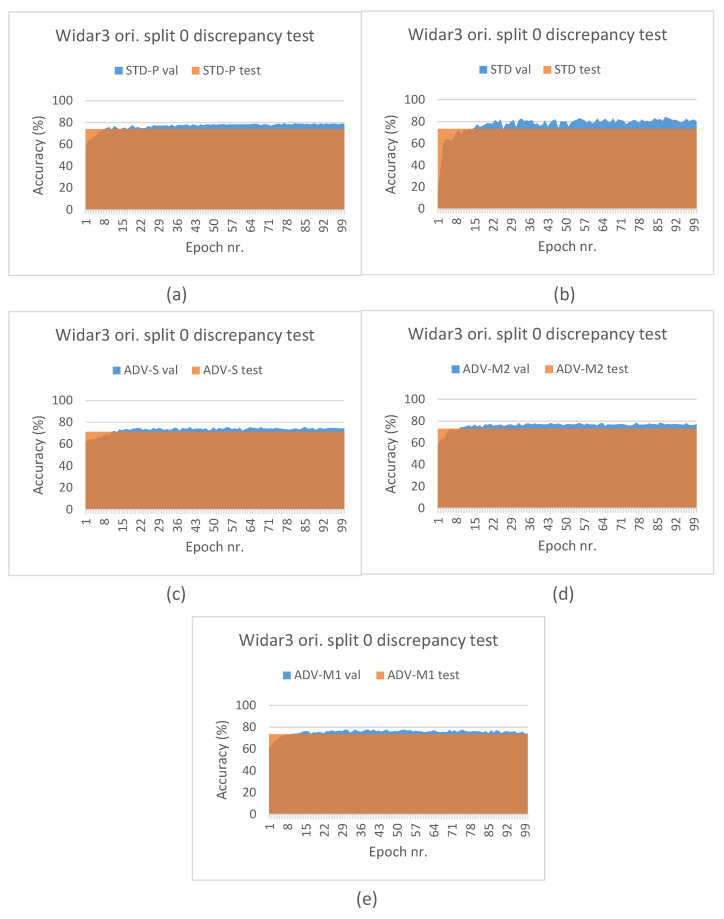
Widar3 domain classification performance discrepancy results when leaving out orientation 1 (orientation leave-out cross-validation split 0), across (**a**) STD-P, (**b**) STD, (**c**) ADV-S, (**d**) ADV-M2, and (**e**) ADV-M1 pipelines.

**Figure 6 sensors-23-09233-f006:**
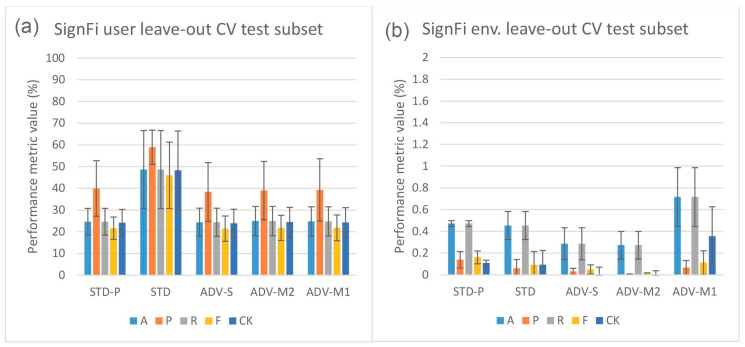
SignFi domain classification (**a**) user and (**b**) environment leave-one-out Cross-Validation (CV) results.

**Figure 7 sensors-23-09233-f007:**
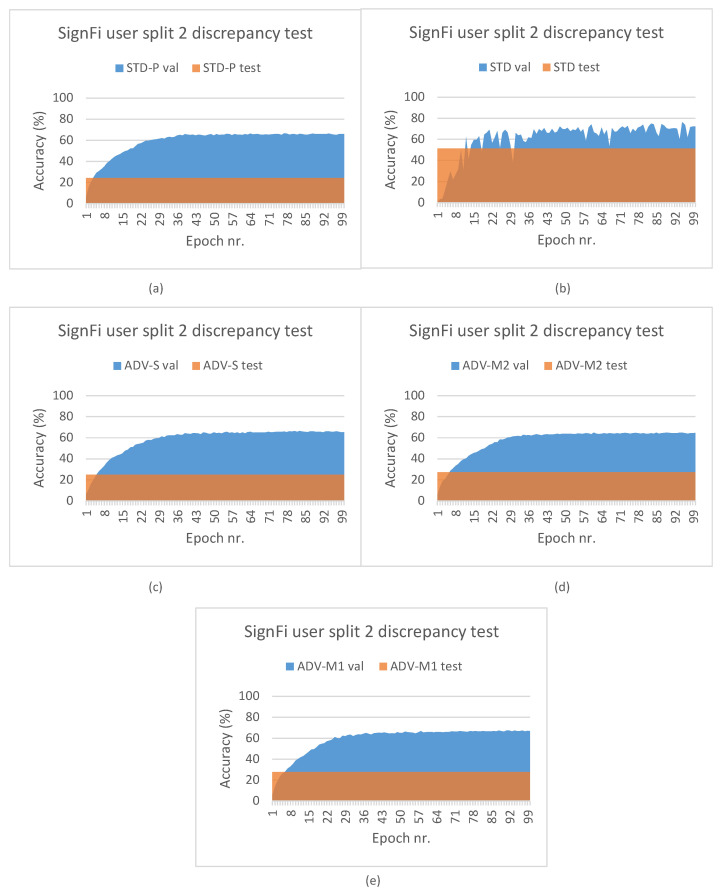
SignFi domain classification performance discrepancy results when leaving out user 3 (under user leave-out cross-validation split 2) across (**a**) STD-P, (**b**) STD, (**c**) ADV-S, (**d**) ADV-M2, and (**e**) ADV-M1 pipelines.

**Figure 8 sensors-23-09233-f008:**
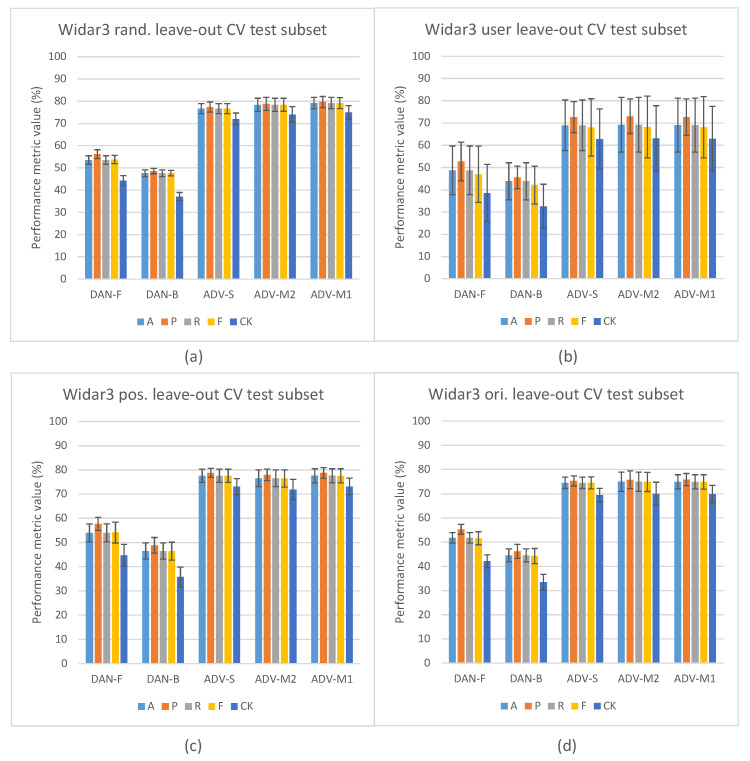
Widar3 domain classification comparison against domain-aware NTXENT pipelines. (**a**) in-domain and (**b**) user, (**c**) position, and (**d**) orientation leave-one-out Cross-Validation (CV) results.

**Figure 9 sensors-23-09233-f009:**
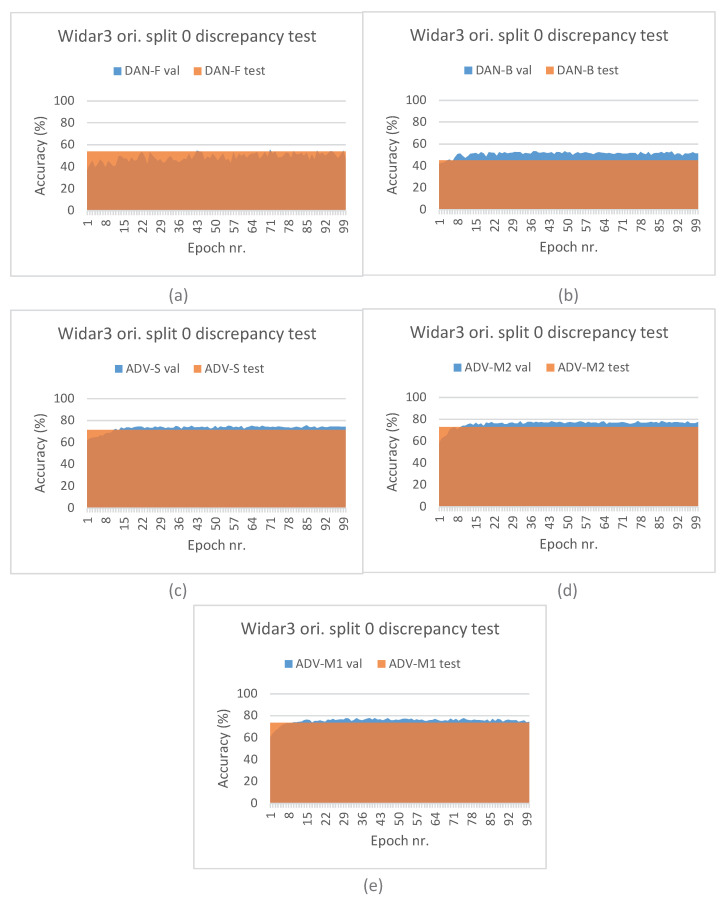
Widar3 domain classification comparison against domain-aware NTXENT pipelines performance discrepancy results when leaving out orientation 1 (under orientation leave-out cross-validation split 0) across (**a**) STD-P, (**b**) STD, (**c**) ADV-S, (**d**) ADV-M2, and (**e**) ADV-M1 pipelines.

**Figure 10 sensors-23-09233-f010:**
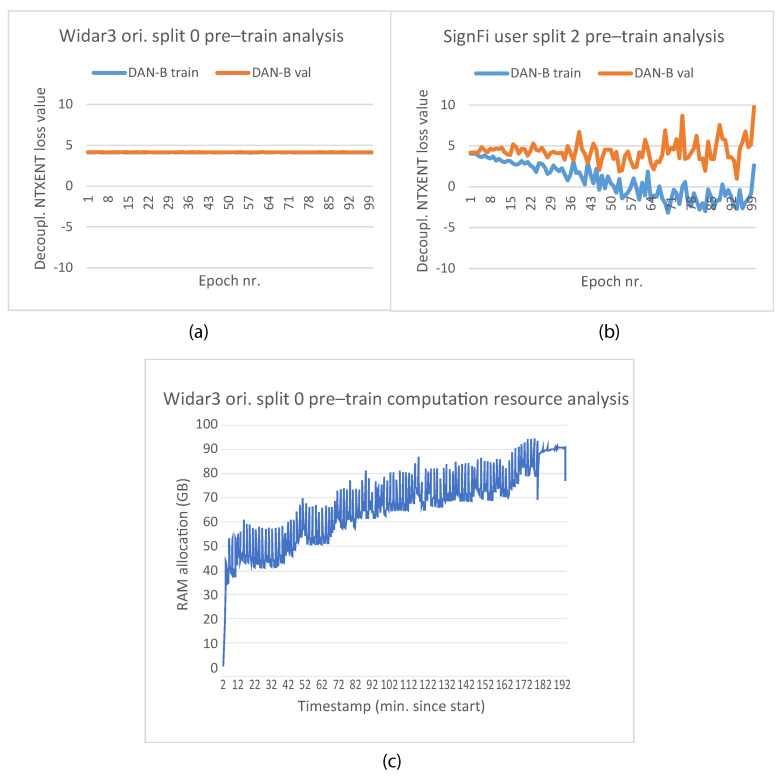
(**a**) Widar3 and (**b**) SignFi domain factor leave-out cross-validation pre-train NTXENT loss results when leaving out orientation 1 (Widar3) or user 3 (SignFi), and (**c**) Widar3 computation resource analysis during aformentioned NTXENT loss result collection for DAN-B pipeline.

**Figure 11 sensors-23-09233-f011:**
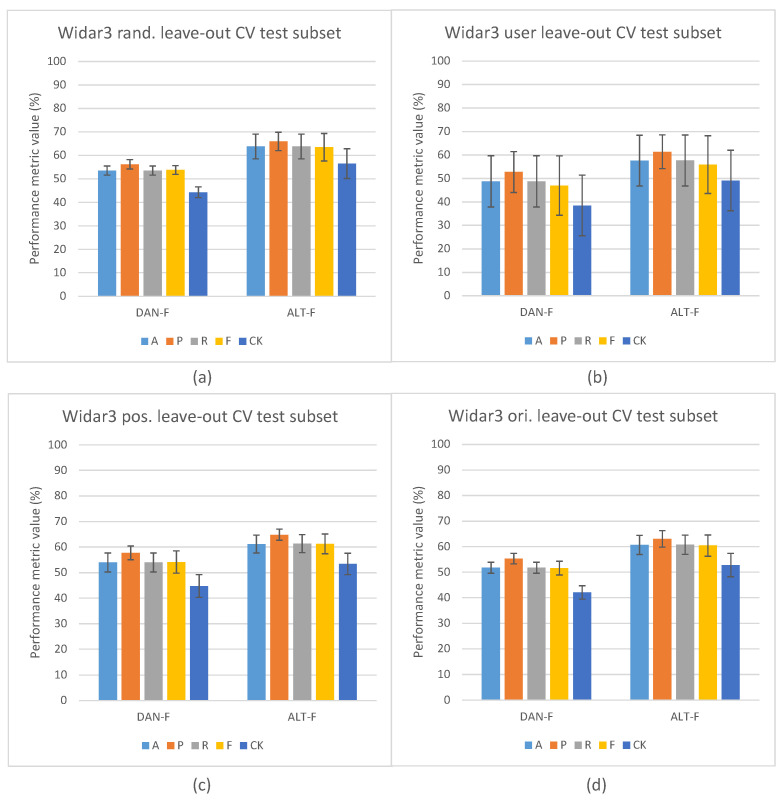
Widar3 domain-aware NTXENT loss comparison between subsequent pre-training/fine-tuning procedures and alternating aformentioned procedures. (**a**) in-domain, (**b**) user, (**c**) position, and (**d**) orientation leave-one-out Cross-Validation (CV) results.

**Figure 12 sensors-23-09233-f012:**
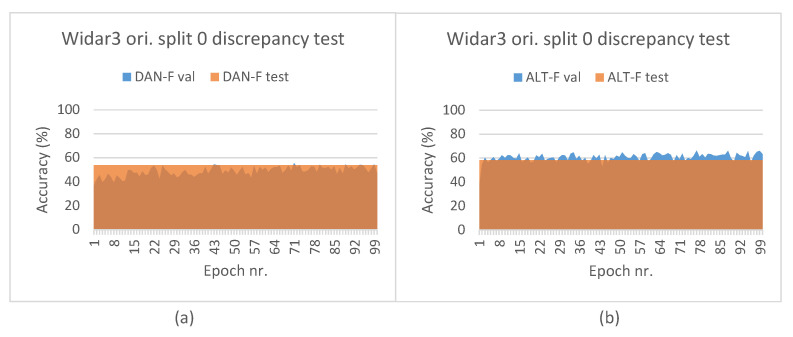
Widar3 domain-aware NTXENT loss comparison between subsequent pre-training/fine-tuning procedures and alternating aforementioned procedures performance discrepancy results when leaving out orientation 1 (under orientation leave-one-out cross-validation split 0) across (**a**) DAN-F and (**b**) ALT-F pipelines.

**Figure 13 sensors-23-09233-f013:**
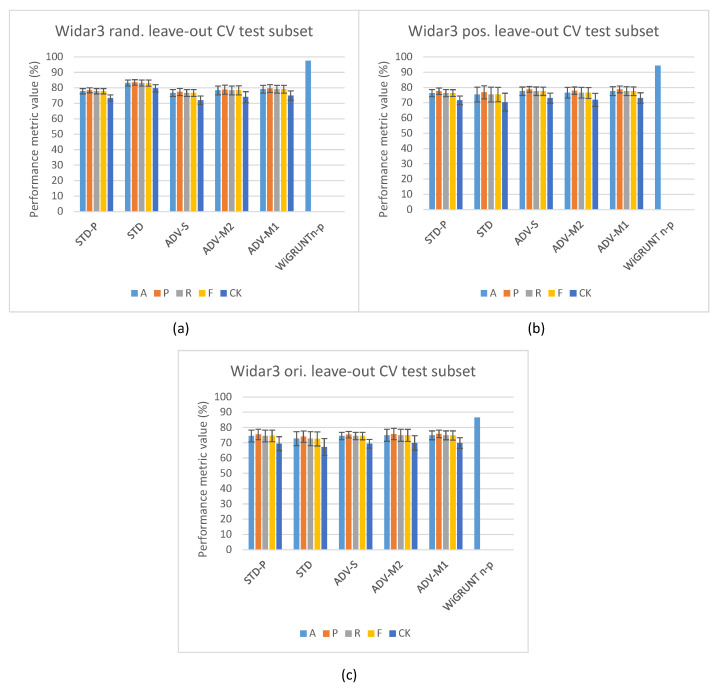
Widar3 domain classification (**a**) in-domain, (**b**) position, and (**c**) orientation leave-one-out Cross-Validation (CV) result comparison against WiGRUNT [22].

**Table 1 sensors-23-09233-t001:** Feature extractor hyperparameter table.

I	Op	KP	Ot	E	Ac	St	B	Sk
224 × 64 × 3	Conv.	12 × 36	21	-	ReLU	1 × 1	✓	-
224 × 64 × 21	Avg. Pool	4 × 16	-	-	-	DP × 2	-	-
112 × 32 × 21	MobileV2	4 × 4	42	2	ReLU	1 × 1	✓	-
112 × 32 × 42	MobileV2	4 × 4	42	2	ReLU	1 × 1	✓	-
112 × 32 × 42	MobileV2	4 × 4	42	2	ReLU	1 × 1	✓	-
112 × 32 × 42	Avg. Pool	26 × 20	-	-	-	DPx2	-	-
56 × 16 × 42	MobileV2	8 × 8	84	4	ReLU	1 × 1	✓	✓
56 × 16 × 84	MobileV2	8 × 8	84	4	ReLU	1 × 1	✓	✓
56 × 16 × 84	Avg. Pool	30 × 22	-	-	-	DP × 2	-	-
28 × 8 × 84	MobileV2	5 × 5	168	7	ReLU	1 × 1	✓	✓
28 × 8 × 168	Avg. Pool	2 × 6	-	-	-	$\frac{I[0]}{2}$× 4	-	-
2 × 2 × 168	Flatten	-	-	-	-	-	-	-

Table denotes input dimension (I), operator type (Op), kernel/pool size (KP), nr. output filters (Ot), expansion rate (E), activation function type (Ac), stride (St), if batch normalization is considered (B), and if a skip connection (Sk) is considered. Timeframe size (T) is 2048 for widar3 subset and 224 for SignFi dataset. Tranceiver link antenna size (A) is 12 for widar3 subset and 3 for SignFi dataset, unless stated otherwise in Section 3.1 and Section 4.3. SignFi example can be found inside the table. Dynamic pool (DP) size is determined via linear function DP=⌊0.00109649·T+1.75439⌋.

**Table 2 sensors-23-09233-t002:** Pipeline -used computation resources and average required wall-clock time per epoch table.

Res.	Data	STD	STD-P	ADV-S	ADV-M1	ADV-M2	DAN-F	ALT-F	DAN-B	ALT-B
CPU	Widar3									
nr.	Widar3	1	1	1	1	1	1	1	1	x
type	Widar3	7R32	7R32	7R32	7R32	7R32	7R32	7R32	7R32	x
cores	Widar3	8	8	8	8	8	8	48	48	x
GPU	Widar3									
nr.	Widar3	1	1	1	1	1	1	4	4	x
type	Widar3	A10G	A10G	A10G	A10G	A10G	A10G	A10G	A10G	x
cores	Widar3	9216	9216	9216	9216	9216	9216	9216	9216	x
mem.	Widar3	24 GB	24 GB	24 GB	24 GB	24 GB	24 GB	24 GB	24 GB	x
RAM	Widar3									
size	Widar3	64 GB	64 GB	64 GB	64 GB	64 GB	64 GB	384 GB	384 GB	x
type	Widar3	DDR5	DDR5	DDR5	DDR5	DDR5	DDR5	DDR5	DDR5	x
acc.	Widar3	69.2 GBs	69.2 GBs	69.2 GBs	69.2 GBs	69.2 GBs	69.2 GBs	69.2 GBs	69.2 GBs	x
SSD	Widar3									
size	Widar3	600 GB	600 GB	600 GB	600 GB	600 GB	600 GB	3.8 TB	3.8 TB	x
int.	Widar3	NVMe	NVMe	NVMe	NVMe	NVMe	NVMe	NVMe	NVMe	x
IOPS	Widar3	20,000	20,000	20,000	20,000	20,000	20,000	80,000	80,000	x
TIME	Widar3	53.1 s	53.4 s	53.8 s	56.5 s	56.6 s	53.3 s	99.8 s	104.4 s	x
CPU	SignFi									
nr.	SignFi	1	1	1	1	1	1	1	1	1
type	SignFi	7R32	7R32	7R32	7R32	7R32	7R32	7R32	7R32	7R32
cores	SignFi	8	8	8	8	8	8	8	8	8
GPU	SignFi									
nr.	SignFi	1	1	1	1	1	1	1	1	1
type	SignFi	A10G	A10G	A10G	A10G	A10G	A10G	A10G	A10G	A10G
cores	SignFi	9216	9216	9216	9216	9216	9216	9216	9216	9216
mem.	SignFi	24 GB	24 GB	24 GB	24 GB	24 GB	24 GB	24 GB	24 GB	24 GB
RAM	SignFi									
size	SignFi	64 GB	64 GB	64 GB	64 GB	64 GB	64 GB	64 GB	64 GB	64 GB
type	SignFi	DDR5	DDR5	DDR5	DDR5	DDR5	DDR5	DDR5	DDR5	DDR5
acc.	SignFi	69.2 GBs	69.2 GBs	69.2 GBs	69.2 GBs	69.2 GBs	69.2 GBs	69.2 GBs	69.2 GBs	69.2 GBs
SSD	SignFi									
size	SignFi	600 GB	600 GB	600 GB	600 GB	600 GB	600 GB	600 GB	600 GB	600 GB
int.	SignFi	NVMe	NVMe	NVMe	NVMe	NVMe	NVMe	NVMe	NVMe	NVMe
IOPS	SignFi	20,000	20,000	20,000	20,000	20,000	20,000	20,000	20,000	20,000
TIME	SignFi	7.4 s	12.9 s	13.7 s	15.6 s	15.6 s	12.9 s	19.4 s	18 s	24 s

Table denotes computation resources and average wall-clock time per epoch for the end-to-end no domain shift mitigation (STD), pre-training no domain shift mitigation (STD-P), standard (ADV-S) domain classification, multi-label domain classification via addition (ADV-M1) or concatenation (ADV-M2), pre-training domain-aware filter ntxent (DAN-F), semi-supervised domain-aware filter ntxent by alternating between pre-train and fine-tune path (ALT-F), pre-training domain-aware batch ntxent (DAN-B), and semi-supervised domain-aware batch ntxent by alternating between pre-train and fine-tune path (ALT-B) pipelines.

## Data Availability

The Widar3.0 dataset can be accessed via IEEE DataPort [accessed 5 May 2023] (https://ieee-dataport.org/open-access/widar-30-wifi-based-activity-recognition-dataset). Links to the SignFi dataset can be found at Github page [accessed 5 May 2023] (https://github.com/yongsen/SignFi).

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
