# Peer review of "Use of Domain Labels during Pre-Training for Domain-Independent WiFi-CSI Gesture Recognition"

_sensors, 2023, doi:10.3390/s23229233_

Round 1

Reviewer 1 Report

Comments and Suggestions for Authors

I found the paper quite difficult to read, in particular because of the extreme verbosity, as can be clearly seen in the abstract (1 page) and the conclusions (2 pages). In the central sections authors are introducing so many different concepts in a raw that is quite impossible to appreciate a systematic methodology. The same for the results, there are more than 35 figures and I could not really understood in the end if results were good or not, not even after reading the conclusions that, by the way, are not concluding, they introduce even more discussions. Conversely, the related work section is very short and does not report recent papers.

- Abstract is way too long, it must be at least reduced 50%. Some comment for the conclusions: first, they should conclude the paper and "draw" conclusions rather than adding more discussions; second, conclusions should be approximately the same size of the abstract. At the moment conclusions are a brief paper on their own.

- List of contributions is too long, please make it shorter (less verbose, it's a list)

- In 3.1 when introducing the dataset domain it's is not clear the meaning of A "transceiver link antenna". As in case of CSI captured on Nss>1 packet there could be notion of both Tx and Rx antennas, clarify the meaning of "transceiver", is it the receiver antenna? Also clarify how many antennas are present in the considered datasets.

- When describing the pre-processing on the two considered datasets it is not clear what is the rational behind the choice of both filter types and filter parameters (i.e., the cut-off frequencies). Apart from that, authors do not clarify m

any of the algorithms they ran on the dataset for removing CFO, CPO etc.

- Figure 6(b) maybe the range of Y axis can be modified as currently all values cannot be understood.

- Fix figures with DAN-F val and DAN-F test as in many of them ony of the two results completely hide the other.

Reviewer 2 Report

Comments and Suggestions for Authors

This paper proposes integration of the adversarial domain classifier in the pre-training phase. It considers this as an effective step towards automatic domain discovery during pre-training. Simulation results show the effectiveness of the proposed algorithm. This paper has innovations, but still has some disadvantages.

1.     The abstract should be refined

2.     The results of Figure 6 (b) are not clear enough.

Comments on the Quality of English Language

It is easy to read.

Reviewer 3 Report

Comments and Suggestions for Authors

1. The authors should clearly describe related work in more detail, contrasting the limitations of the related works. Moreover, the reviewer recommend to ease the overview related works by using overview tables.

2. The proposed processes should be revised in a more formal pseudocode template. The authors should include more technical details and explanations.

3. Some parameters and their values are unknown. It would be better to show all these parameters and explain the reason for those numbers in the table.

4. How about the computation complexity of the proposed method compared with related work? The performance comparison to other improved schemes is required, such as [A].

5. The experiment results show the performance with high accuracy, please show the parameter settings of each approach using a table.

6. The discussion section in the present form is relatively weak and should be strengthened with more details and justifications.

[A] A. Zinys, B. Berlo and N. Meratnia, A Domain-Independent Generative Adversarial Network for Activity Recognition Using WiFi CSI Data, Sensors, vol. 21, no. 23, pp. 1-23, 2021.

Round 2

Reviewer 1 Report

Comments and Suggestions for Authors

Authors worked out the paper according to my previous suggestions. I am satisfied with the current document.

Reviewer 3 Report

Comments and Suggestions for Authors

This paper has edited and revised according to the reviewer's suggestions.